# From Atoms to Chains: Divergence-Guided Reasoning Curriculum for Unlabeled LLM Domain Adaptation

## Abstract

Adapting Large Language Models (LLMs) to specialized domains without human-annotated data is a crucial yet formidable challenge. Widely adopted knowledge distillation methods often devolve into coarse-grained mimicry, where the student model inefficiently targets its own weaknesses and risks inheriting the teacher's reasoning flaws. This exposes a critical pedagogical dilemma: how to devise a reliable curriculum when the teacher itself is not an infallible expert. Our work resolves this by capitalizing on a key insight: while LLMs may exhibit fallibility in complex, holistic reasoning, they often exhibit high fidelity on focused, atomic sub-problems. Based on this, we propose **D**ivergence-**G**uided **R**easoning Curriculum (DGRC), which constructs a learning path from atomic knowledge to reasoning chains by dynamically deriving two complementary curricula from disagreements in reasoning pathways. When a student and teacher produce conflicting results, DGRC directs the teacher to perform a diagnostic analysis: it analyzes both reasoning paths to formulate atomic queries that target the specific points of divergence, and then self-answers these queries to create high-confidence atomic question-answer pairs. These pairs then serve a dual purpose: (1) providing an **atomic curriculum** to rectify the student's knowledge gaps, and (2) serving as factual criteria to filter the teacher's original reasoning chains, yielding a **verified CoT curriculum** that teaches the student how to integrate atomic knowledge into complete reasoning paths. Experiments across the medical and legal domains on student models of various sizes demonstrate the effectiveness of our DGRC framework. Notably, our method achieves a 7.76% relative improvement for the 1.5B student model in the medical domain over strong unlabeled baseline.

## 1 Introduction

Large Language Models (LLMs) have established a strong performance baseline by mastering foundational linguistic and factual knowledge (Grattafiori et al., 2024; Team et al., 2024; Guo et al., 2025a; Yang et al., 2025). However, advancing from this foundational capability to expert-level reasoning in specialized domains is a significant leap that requires targeted adaptation (Wang et al., 2023; Iacovides et al., 2024). This process remains a formidable challenge, primarily due to the prohibitive cost and effort of curating high-quality, human-annotated datasets. Consequently, developing effective methods for unlabeled LLM adaptation has become a critical frontier.

Research within this frontier has largely advanced along two main avenues. The first is LLM knowledge distillation (Taori et al., 2023; Hsieh et al., 2023; Xu et al., 2024), which involves a student model mimicking the reasoning process of a more capable teacher. However, this approach confronts two fundamental challenges in the unlabeled setting: (1) the mimicry is too coarse-grained to efficiently address the student's specific weaknesses, and (2) the teacher's supervision is inherently unreliable without ground-truth labels, risking the propagation of its own reasoning flaws. The second paradigm attempts to bypass the teacher's fallibility by generating training data from external knowledge bases (Zhang et al., 2024; Qin et al., 2025; Guo et al., 2025b), but this strategy is often constrained by the high cost of curating domain-specific resources.

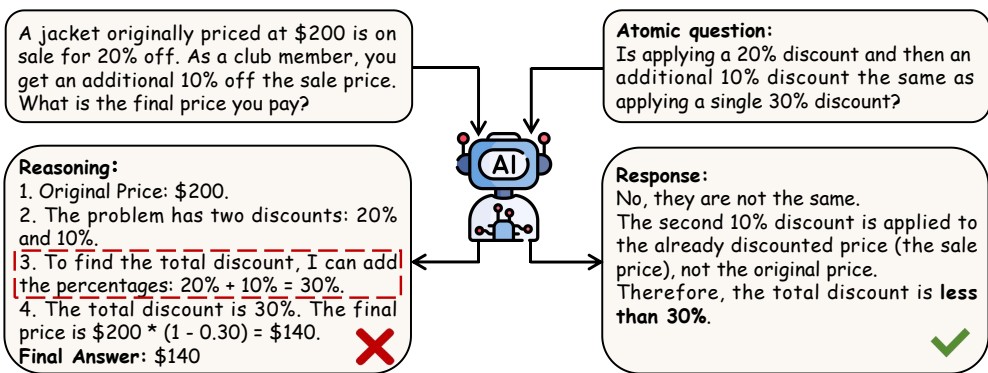

Figure 1: An illustration of the cognitive asymmetry in LLMs. The same LLM exhibits fallibility on a multi-step problem by incorrectly applying a flawed heuristic (left), yet demonstrates high fidelity when prompted with an atomic question that isolates the core conceptual error (right).

These limitations expose a critical pedagogical dilemma: how to devise a reliable curriculum without resorting to costly external knowledge bases when the teacher model itself is not an infallible expert. Our work resolves this dilemma by leveraging a key insight we term **cognitive asymmetry**: while LLMs often fail at complex, holistic reasoning, their performance on focused, atomic sub-problems is highly reliable. This principle is supported by extensive research in problem decomposition (Zhou et al., 2022; Xue et al., 2024; Zhou et al., 2025) and is validated by our experiments in Section 4.3. Figure 1 illustrates this phenomenon. The same LLM that incorrectly applies a flawed heuristic (conflating sequential discounts) in a full Chain-of-Thought process can accurately articulate the correct principle when prompted with a focused question.

Grounded in this insight, we propose the **D**ivergence-**G**uided **R**easoning **C**urriculum (DGRC), a framework that dynamically generates a two-part curriculum based on model disagreements to guide a student's learning path from foundational atomic knowledge to complex reasoning chains, which redefines the teacher-student interaction. Instead of assuming the student is at fault when results conflict, DGRC treats the divergence as a trigger for an impartial inquiry. The teacher model is tasked to act as a neutral diagnostician: it analyzes both reasoning paths to formulate atomic, root-cause queries about the precise point of divergence. Crucially, the teacher answers these focused queries independently of its original reasoning chain, capitalizing on its high fidelity for such atomic tasks. After a quality filtering stage, the generated high-confidence atomic question-answer pairs serve a dual purpose that directly resolves the initial pedagogical dilemma: (1) they constitute the **atomic curriculum**, used to rectify the student's foundational knowledge gaps, replacing coarse-grained mimicry with targeted lessons, and (2) they act as factual criteria to filter the teacher's original thought process, mitigating the propagation of flaws and yielding a **verified CoT curriculum** for training the student to compose complex reasoning chains.

To comprehensively validate our approach, we conduct extensive experiments across a wide range of specialized domains, models, and training paradigms. Our results demonstrate that DGRC is a highly versatile framework that delivers consistent and substantial performance improvements across different teacher-student pairs (Section 4.4), in a self-teaching configuration (Section 4.5), and with various training strategies (Section 4.7). Notably, our approach is particularly effective on smaller-scale models, boosting the performance of a 1.5B student model by a relative 7.76% in the medical domain compared to a strong unlabeled baseline.

In summary, our main contributions are threefold:

- A novel self-supervised framework, DGRC, that enhances the complex reasoning abilities of LLMs in specialized domains without requiring any human-annotated data.

- A novel mechanism that automatically transforms reasoning divergence into a dual-curriculum learning path: an atomic curriculum to rectify factual knowledge and a verified CoT curriculum to master compositional reasoning.

- Extensive empirical validation demonstrating that DGRC achieves substantial performance gains across diverse domains and models, particularly boosting the capabilities of smaller models.

## 2  RELATED WORK

**Problem Decomposition in LLMs.**  Decomposing complex problems into simpler, manageable sub-problems is a cornerstone strategy for enhancing reasoning in LLMs. This approach stems from an insight we term the cognitive asymmetry of LLMs: a powerful model prone to errors on a complex task can reliably solve its constituent atomic steps. This insight has driven the evolution of methods, from implicit step-by-step reasoning (Wei et al., 2022; Feng et al., 2023), to explicit sub-task decomposition (Zhou et al., 2022; Xue et al., 2024; Zhou et al., 2025), and graph-based exploration of multiple reasoning paths (Yao et al., 2023; Besta et al., 2024). While these methods have validated the value of decomposition, their strategies typically depend on manually engineered exemplars or fixed heuristics. In contrast, DGRC introduces a novel, data-driven paradigm that utilizes reasoning divergence as an automatic signal to trigger and pinpoint the precise atomic questions that need to be resolved.

**LLM Adaptation in Unlabeled Settings.**  The challenge of adapting LLMs to specialized domains without annotated data has spawned two main paradigms. The first, LLM Knowledge Distillation, trains a student model to mimic the reasoning of a more capable teacher (Taori et al., 2023; Hsieh et al., 2023). However, this approach is fundamentally hampered by the coarse-grained nature of the mimicry and the inherent fallibility of the teacher in unlabeled settings. The second paradigm attempts to circumvent this by synthesizing training data from external knowledge bases (Zhang et al., 2024; Qin et al., 2025; Guo et al., 2025b). However, this strategy is often constrained by the high cost of curating such external resources. This highlights the need for new adaptation methods that can generate fine-grained curriculum from the models' internal reasoning, thereby overcoming the limitations of both coarse-grained mimicry and the high cost of external resources.

**Learning from Model Feedback.**  Our work is also relevant to the broader field of learning from model feedback. Many existing approaches can be categorized as learning from error, such as self-correction, where an LLM revises its own output (Madaan et al., 2023; Song et al., 2025; Alazraki et al., 2025), and process supervision, which uses a verification model to score each reasoning step (Lightman et al., 2023). However, a core challenge for these methods in unlabeled settings is the difficulty of reliably identifying an error without ground-truth labels. Our method sidesteps this challenge by shifting the paradigm from learning from error to learning from disagreement. Instead of requiring a clear error signal, it uses the readily available divergence between two models to pinpoint their points of uncertainty and transform them into targeted learning opportunities.

## 3  DIVERGENCE-GUIDED REASONING CURRICULUM

### 3.1  OVERVIEW

This work addresses the challenge of adapting a student Large Language Model (LLM) to a specialized domain using only a set of unlabeled problems. Formally, given a collection of unlabeled problems $\mathcal{D}_{\text{unlabeled}} = \{p_1, p_2, ..., p_N\}$, a powerful teacher model $\mathcal{M}_T$, and a student model $\mathcal{M}_S$, our goal is to enhance the student's domain-specific reasoning capabilities without access to ground-truth solutions.

To this end, we propose the Divergence-Guided Reasoning Curriculum (DGRC), a multi-stage framework that harnesses reasoning discrepancies between teacher and student models to construct a structured learning path. As illustrated in Figure 2, DGRC operates in three consecutive stages. The process begins with the **divergence detection** stage, where disagreements in the models' final answers trigger the learning process. Triggered by these disagreements, the **curriculum generation** stage produces two complementary curricula: an atomic curriculum designed to address the root causes of divergence, and a verified CoT curriculum comprising filtered, high-quality reasoning chains from the teacher. The framework concludes with the **student adaptation** stage, where the student model is trained on these curricula, implementing an "atom-to-chain" learning paradigm to systematically enhance its reasoning abilities.

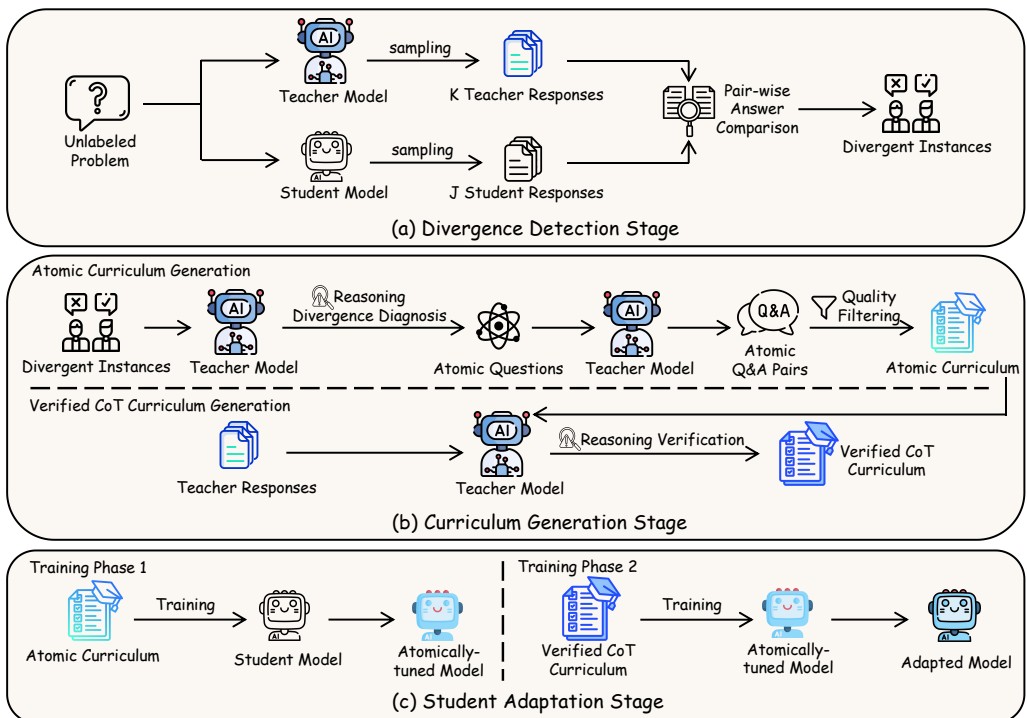

Figure 2: The overview of the Divergence-Guided Reasoning Curriculum (DGRC) framework.

## 3.2 DIVERGENCE DETECTION STAGE

The DGRC framework initiates with the divergence detection stage, as illustrated in Figure 2 (a). The goal of this stage is to efficiently generate a pool of divergent instances by sampling multiple responses from the teacher and student models and performing a pair-wise comparison of their answers.

Specifically, for each unlabeled problem $p^i \in \mathcal{D}_{\text{unlabeled}}$, we sample $K$ responses from the teacher and $J$ from the student, yielding the respective output sets $\mathcal{O}_T^i = \{o_{T,1}^i, ..., o_{T,K}^i\}$ and $\mathcal{O}_S^i = \{o_{S,1}^i, ..., o_{S,J}^i\}$. Each response $o^i = (c^i, a^i)$ contains a reasoning chain $c^i$ and a final answer $a^i$.

We then conduct a systematic pair-wise comparison, evaluating each of the $K$ teacher responses against each of the $J$ student responses. A pair is flagged as divergent if their final answers conflict (see Appendix A for specific criteria). Formally, this produces a set of divergent pairs for each problem $p^i$:

$$D_{p^i} = \{(o_T^i, o_S^i) | o_T^i \in \mathcal{O}_T^i, o_S^i \in \mathcal{O}_S^i, \text{and } a_T^i \neq a_S^i\}. \tag{1}$$

This stage culminates in a diagnostic dataset, $\mathcal{D}_{\text{diag}}$, which aggregates all problems exhibiting at least one divergence:

$$D_{\text{diag}} = \{(p^i, D_{p^i}, \mathcal{O}_T^i) | p^i \in \mathcal{D}_{\text{unlabeled}}, \text{and } D_{p^i} \neq \emptyset\}. \tag{2}$$

Each entry in this dataset encapsulates the necessary information for the subsequent stages: the divergent pairs ($D_{p^i}$) are passed on for atomic curriculum generation stage, while the set of teacher responses ($\mathcal{O}_T^i$) is retained for the verified CoT curriculum generation stage.

## 3.3 CURRICULUM GENERATION STAGE

Upon identifying divergent instances in $D_{\text{diag}}$, the framework proceeds to its conceptual core: the curriculum generation stage, depicted in Figure 2 (b). In this stage, abstract model disagreements are transformed into two concrete, complementary curricula through two sequential sub-stages.

### 3.3.1 Atomic Curriculum Generation

This sub-stage diagnoses the root causes of reasoning divergence and distills them into an atomic curriculum. The process begins by iterating through each problem $p^i$ in the diagnostic set $D_{\text{diag}}$.

For each divergent instance $(o_T^i, o_S^i) \in D_{p^i}$, we prompt the teacher model to act as a diagnostician, analyzing both its own reasoning $c_T^i$ and the student's $c_S^i$. This analysis, guided by a carefully engineered prompt (Appendix B.1), synthesizes a set of atomic questions. Each question is a fundamental, self-contained query designed to isolate a single point of factual or logical discrepancy between the two chains.

The teacher then answers each atomic question de novo, unconstrained by its original reasoning. This step leverages the cognitive asymmetry principle: a model's accuracy on focused, atomic queries typically surpasses its reliability on complex, multi-step tasks. After a quality filtering process (Appendix C), the resulting high-confidence Q&A pairs—constituting verified atomic knowledge—are aggregated to form the atomic curriculum.

### 3.3.2 Verified CoT Curriculum Generation

This sub-stage curates the verified CoT curriculum by using the previously generated atomic knowledge for quality control. For each problem $p^i \in D_{\text{diag}}$, we consolidate its verified, high-confidence atomic Q&A pairs into a verification set, denoted as $\mathcal{A}_i$. This set then serves as a factual standard to audit the teacher's original reasoning chains ($\mathcal{O}_T^i$). Using a dedicated prompt (Appendix B.2), the teacher model assesses each chain $c_{T,k}^i$ against $\mathcal{A}_i$, retaining only those that exhibit no factual inconsistencies. This rigorous auditing process yields the verified CoT curriculum: a collection of high-fidelity, multi-step reasoning exemplars.

### 3.4 Student Adaptation Stage

As illustrated in Figure 2 (c), the student adaptation stage implements our "atom-to-chain" learning philosophy through a two-phase training process. First, the student model is fine-tuned on the atomic curriculum, a crucial step that rectifies foundational knowledge gaps to produce an intermediate, atomically-tuned model. This improved model is then trained on the verified CoT curriculum, leveraging its high-fidelity exemplars to master the composition of complex reasoning chains.

## 4 Experiments

Our comprehensive experiments validate the effectiveness and versatility of the DGRC framework. We first establish its superior performance against strong baselines across two specialized domains (Section 4.2). We then demonstrate its flexibility across various model configurations (Section 4.4), in a self-teaching setting (Section 4.5), and when integrated with reinforcement learning to probe its upper performance limits (Section 4.7).

To dissect the framework's core mechanisms, we conduct a series of in-depth analyses. We empirically validate our foundational "cognitive asymmetry" insight (Section 4.3) and perform crucial ablation studies on the curriculum components (Section 4.6). Further detailed analyses in the Appendix quantify the effectiveness of our CoT verification step (Appendix H), study the impact of sampling counts (Appendix J), and provide a qualitative analysis that links model uncertainty to the errors targeted by our method (Appendix K).

### 4.1 Experimental Setup

**Domains and Benchmarks.** We evaluate DGRC's effectiveness across two specialized domains: medical and legal. For the medical domain, we provide a comprehensive assessment by testing the adapted model on three distinct benchmarks: the validation set of MedMCQA (Pal et al., 2022), the test set of MedQA-USMLE (Jin et al., 2021), and the professional medicine subset of the MMLU benchmark (Hendrycks et al., 2020). For the legal domain, evaluation is conducted on the validation set of CaseHOLD (Zheng et al., 2021) and the professional law subset of MMLU.

**Training Data.** The DGRC curricula for each domain are generated using a pool of corresponding unlabeled queries. For the medical domain, we utilize 182,822 questions from the official training set of MedMCQA (Pal et al., 2022). For the legal domain, the curriculum is generated from 42,509 case problems in the CaseHOLD (Zheng et al., 2021) training set. All ground-truth labels and answers are discarded to strictly maintain our unlabeled adaptation setting. Detailed statistics on the data volumes are provided in Appendix G.

**Models.** To evaluate the versatility and scalability of DGRC, we employ a multi-teacher, multi-student configuration. We use three distinct teacher models to generate the curricula, spanning both proprietary model (GPT-4.1 (Achiam et al., 2023)) and powerful open-source options (Qwen2.5-Instruct-32B and -72B (An et al., 2024)). The student models targeted for adaptation are three smaller-scale versions from the same family: Qwen2.5-Instruct-1.5B, -3B, and -7B. Our main results are reported using GPT-4.1 as the teacher, and we validate the framework's generalizability with open-source teachers in Section 4.4.

**Baselines.** We compare DGRC against three strong adaptation baselines. The first, Baseline-w/o-label, is an unlabeled competitor that trains on the reasoning chain corresponding to the majority final answer among $K$ teacher-generated samples. The second, Baseline-w/-label, acts as a supervised upper bound by using ground-truth labels to select only correct reasoning chains for training. The third, RLAIF Lee et al. (2023) (GRPO), acts as a reinforcement learning baseline where the student model is optimized using Group Relative Policy Optimization (Shao et al., 2024). In this setting, the teacher model serves as a reward model to provide scalar feedback on the correctness of the student's reasoning and final answer.

**Implementation Details.** To ensure a fair comparison, our main experiments implement DGRC and the distillation baselines using Supervised Fine-Tuning (SFT), while the RLAIF baseline follows the standard RL training protocol. However, the DGRC framework itself is compatible with various training paradigms, a point we explore further in Section 4.7. More implementation details are provided in Appendix F.

## 4.2 MAIN RESULTS

Our main results are summarized in Table 1, which presents a comprehensive performance comparison across four distinct categories of models on benchmarks in the medical and legal domains. These categories include: (1) leading proprietary models, including GPT-4.1 (Achiam et al., 2023) and Gemini-2.5-Pro (Comanici et al., 2025); (2) general-purpose open-source models, including the Qwen2.5 series (An et al., 2024), the Llama3.1 series Grattafiori et al. (2024), and DeepSeek-R1 (Guo et al., 2025a); (3) specialized domain-expert models including the MegaScience series (Fan et al., 2025), LegalHal (Hu et al., 2025) MediTron (Chen et al., 2023), and Med-PaLM2 (Singhal et al., 2025); and (4) our proposed DGRC method alongside three strong baselines (Supervised Distillation, Unlabeled Distillation, and RLAIF), evaluated at various model scales. From these results, we highlight four key findings.

**DGRC Outperforms Distillation Baselines.** Across all model scales, DGRC consistently and substantially outperforms the unlabeled distillation baseline. This is exemplified in the medical domain, where the 1.5B student achieves a 7.76% relative improvement over the Baseline-w/o-label. Perhaps most strikingly, in the medical domain, the DGRC-adapted 3B model even surpasses the Baseline-w/-label with a 3.71% relative gain. This result is significant: it suggests that DGRC's method of identifying and correcting specific errors provides a more potent learning signal than simply imitating pre-verified reasoning paths. By forcing the model to engage in active repair instead of passive imitation, DGRC fosters a deeper and more robust understanding of the knowledge, even when compared to learning from oracle-selected examples.

**Competitiveness against Reinforcement Learning.** To further validate DGRC's effectiveness, we compare it with Reinforcement Learning from AI Feedback (RLAIF), implemented via Group Relative Policy Optimization (GRPO) using GPT-4.1 as the judge. As shown in Table 1, DGRC (via SFT) achieves competitive or superior performance across scales, demonstrating that our curriculum-based approach can act as a highly effective adaptation method. Furthermore, we note

Table 1: Main results on medical and legal domains. We report accuracy (%) across five benchmarks, with averaged results for the medical (Avg-M) and legal (Avg-L) domains. **Label** indicates if the method requires human-annotated domain-specific training data for the adaptation stage (✓) or not (✗). **Knowledge** indicates if the method requires an external knowledge base (✓) or not (✗). Performance for models marked with † is reported from their respective original publications.

| Model / Method | Params | Label | Knowledge | MedMCQA | MedQA | MMLU-M | Avg-M | CaseHOLD | MMLU-L | Avg-L |
|---|---|---|---|---|---|---|---|---|---|---|
| *Proprietary Models* | | | | | | | | | | |
| GPT-4.1 (25-04-14) | N/A | ✗ | ✗ | 81.3 | 92.5 | 97.3 | 90.4 | 76.4 | 91.7 | 84.1 |
| Gemini-2.5-Pro (25-06-05) | N/A | ✗ | ✗ | 82.1 | 89.7 | 98.1 | 90.0 | 79.0 | 92.6 | 85.8 |
| *General-Purpose Open-Source Models* | | | | | | | | | | |
| Qwen2.5-Instruct | 1.5B | ✗ | ✗ | 37.1 | 40.1 | 54.8 | 44.0 | 44.1 | 61.2 | 52.7 |
| Qwen2.5-Instruct | 3B | ✗ | ✗ | 38.4 | 41.6 | 63.7 | 47.9 | 38.9 | 61.2 | 50.1 |
| Qwen2.5-Instruct | 7B | ✗ | ✗ | 55.6 | 59.8 | 86.6 | 67.3 | 63.9 | 75.2 | 69.6 |
| Qwen2.5-Instruct | 32B | ✗ | ✗ | 63.1 | 73.1 | 93.2 | 76.5 | 68.2 | 84.3 | 76.3 |
| Qwen2.5-Instruct | 72B | ✗ | ✗ | 68.6 | 79.6 | 94.1 | 80.8 | 71.4 | 86.8 | 79.1 |
| Llama3.1-Instruct | 8B | ✗ | ✗ | 56.5 | 65.8 | 80.5 | 67.6 | 57.7 | 76.9 | 67.3 |
| Llama3.1-Instruct | 70B | ✗ | ✗ | 72.2 | 82.3 | 92.4 | 82.3 | 69.9 | 87.6 | 78.8 |
| DeepSeek-R1 | 671B | ✗ | ✗ | 78.7 | 91.0 | 99.0 | 89.6 | 74.6 | 91.7 | 83.2 |
| *Domain Expert Models* | | | | | | | | | | |
| MegaScience | 1.5B | ✗ | ✓ | 36.5 | 31.2 | 61.2 | 43.0 | 40.9 | 63.6 | 52.3 |
| MegaScience | 3B | ✗ | ✓ | 44.3 | 40.0 | 73.3 | 52.5 | 52.4 | 66.1 | 59.3 |
| MegaScience | 7B | ✗ | ✓ | 57.4 | 61.0 | 87.1 | 68.5 | 60.5 | 81.0 | 70.8 |
| LegalHal | 8B | ✗ | ✓ | - | - | - | - | 63.7 | 73.6 | 68.7 |
| MediTron† | 70B | ✓ | ✓ | 53.3 | 59.8 | 71.5 | 61.5 | - | - | - |
| Med-PaLM2† | 340B | ✓ | ✗ | 72.3 | 85.4 | 92.0 | 83.2 | - | - | - |
| *Divergence-Guided Reasoning Curriculum* | | | | | | | | | | |
| Baseline-w/-label | 1.5B | ✓ | ✗ | 49.2 | 49.8 | 72.2 | 57.1 | 73.8 | 65.3 | 69.6 |
| Baseline-w/o-label | 1.5B | ✗ | ✗ | 48.9 | 48.3 | 65.0 | 54.1 | 70.3 | 62.8 | 66.6 |
| RLAIF (GRPO) | 1.5B | ✗ | ✗ | 50.2 | 49.5 | 67.5 | 55.7 | 71.2 | 64.5 | 67.9 |
| DGRC (Ours) | 1.5B | ✗ | ✗ | 51.7 | 50.8 | 72.2 | **58.3** | 71.5 | 67.8 | **69.7** |
| Baseline-w/-label | 3B | ✓ | ✗ | 57.6 | 59.1 | 77.3 | 64.7 | 75.6 | 69.4 | 72.5 |
| Baseline-w/o-label | 3B | ✗ | ✗ | 56.5 | 58.4 | 73.5 | 62.8 | 72.6 | 69.4 | 71.0 |
| RLAIF (GRPO) | 3B | ✗ | ✗ | 58.2 | 60.1 | 76.5 | 64.9 | 73.0 | 70.8 | 71.9 |
| DGRC (Ours) | 3B | ✗ | ✗ | 59.7 | 60.6 | 81.0 | **67.1** | 73.1 | 72.7 | **72.9** |
| Baseline-w/-label | 7B | ✓ | ✗ | 64.5 | 70.1 | 90.1 | 74.9 | 76.9 | 78.5 | **77.7** |
| Baseline-w/o-label | 7B | ✗ | ✗ | 64.3 | 69.4 | 87.2 | 73.7 | 73.6 | 78.5 | 76.1 |
| RLAIF (GRPO) | 7B | ✗ | ✗ | 68.1 | 71.4 | 89.1 | 76.2 | 74.4 | 79.0 | 76.7 |
| DGRC (Ours) | 7B | ✗ | ✗ | 67.5 | 72.8 | 90.9 | **77.1** | 74.3 | 79.3 | 76.8 |

that DGRC is methodologically distinct from RLAIF: DGRC functions as a data synthesis engine that generates high-quality curricula from model divergence, whereas RLAIF is an optimization paradigm. This suggests that while DGRC is competitive on its own, its generated curricula could potentially serve as robust high-quality data to further enhance RLAIF training, a synergy we partially explore in Appendix I.

**DGRC Delivers Superior Generalization.** A key advantage of DGRC is its enhanced generalization capability. Our adaptation training is performed exclusively on the two benchmarks: MedMCQA for medicine and CaseHOLD for law. However, DGRC's performance gains are most pronounced on the unseen benchmarks (MedQA, MMLU-M, and MMLU-L). Crucially, on these generalization tasks, DGRC not only surpasses the unlabeled baseline but also consistently outperforms the Baseline-w/-label. This indicates that by focusing on correcting foundational, atomic errors, DGRC effectively prevents overfitting to the patterns of the training benchmarks. Instead, it compels the model to learn portable atomic knowledge that are applicable across a wider range of unseen tasks.

**DGRC Achieves High Parameter Efficiency.** DGRC demonstrates remarkable parameter efficiency, enabling smaller models to achieve highly competitive performance. While a performance gap remains compared to top proprietary models like GPT-4.1, our DGRC-adapted Qwen2.5-Instruct-7B student successfully surpasses its much larger 32B sibling model. Furthermore, our adapted models outperform several larger domain-expert models, demonstrating that DGRC's general reasoning curriculum can be more effective than specialized pre-training. For instance, our 3B adapted student surpasses the 70B-parameter MediTron, while our 1.5B model outperforms the 8B LegalHal.

## 4.3 VALIDATING COGNITIVE ASYMMETRY AND ATOMIC KNOWLEDGE QUALITY

The core assumption of our work is the principle of cognitive asymmetry. To validate this assumption and to quantitatively measure the quality of the generated atomic knowledge, we de-

Table 2: Peer-correction results and analysis of atomic knowledge quality. **Atomic Q&A Acc. (Human)** is the manual evaluation on 100 sampled pairs. **Atomic Q&A Acc. (Auto)** is the automated evaluation on 2,000 sampled pairs using Gemini-2.5-Pro as the judge. **Peer Reliability** measures the consistency (Pass@5) of the peer model on the divergent problems. **Correction Rate** is the percentage of initial errors a model successfully fixed using its generated atomic knowledge.

| Model | # of Problems | # of Atomic Q&As | Atomic Q&A Acc. (Human, 100 samples) | Atomic Q&A Acc. (Auto, 2k samples) | Peer Reliability (Pass@5) | Correction Rate |
|---|---|---|---|---|---|---|
| GPT-4.1 (25-04-14) | 212 | 843 | 95.0% | 94.2% | 55.4% | 22.6% |
| Qwen2.5-Instruct-72B | 734 | 4,037 | 88.0% | 86.5% | 79.1% | 34.6% |

sign a straightforward peer-correction experiment. The setup for this process is as follows: we first curate two sets of problems from MedMCQA benchmark where our teacher models, GPT-4.1 and Qwen2.5-Instruct-72B, disagree—one succeeding where the other fails. For each problem, the model that initially fails is tasked with a DGRC-style diagnostic process: it analyzes the divergence between its own flawed reasoning and the correct reasoning provided by its peer, generating a set of atomic Q&A pairs.

To assess the quality of atomic knowledge, we employ a dual-evaluation strategy. First, we randomly sample 100 Q&A pairs generated by each model to ensure diversity, followed by rigorous manual evaluation by domain experts. Second, to verify the robustness of our findings on a larger scale, we employ Gemini-2.5-Pro as an automated judge to evaluate a much larger subset of 2,000 atomic Q&A pairs. The specific prompts and criteria used for this automated judgment are detailed inAppendix D. Subsequently, the failing model is prompted to re-evaluate its original reasoning chain against the atomic knowledge it has just created. The specific prompt used for this instruction is detailed in Appendix E. If an inconsistency is found, the model must correct the specific flawed step and then resume the reasoning process from that point forward to generate a new final answer. We then measure the final correction rate based on this revised reasoning.

The results, presented in Table 2, offer two key insights. First, both manual and automated evaluations consistently confirm that our method generates atomic knowledge of exceptionally high quality. Manual evaluation yields accuracy rates of 95.0% for GPT-4.1 and 88.0% for Qwen2.5-Instruct-72B. Crucially, these findings are corroborated by the large-scale automated evaluation, which reports highly consistent accuracy rates of 94.2% and 86.5%, respectively. This cross-validated evidence directly demonstrates that our divergence-guided approach is highly effective at synthesizing reliable, foundational facts, even when the model's initial holistic reasoning is flawed.

Second, we observe an intriguing phenomenon: GPT-4.1 achieves superior atomic accuracy yet a lower final correction rate (22.6%) compared to the Qwen2.5-Instruct-72B student (34.6%). We hypothesize that this discrepancy is driven by the reliability of the peer's reasoning used for diagnosis. In our setup, correction is triggered by divergence: when a model fails, the curriculum is derived from the reasoning of its successful peer. However, a correct final answer does not guarantee correct reasoning—the peer might have "guessed" correctly via flawed heuristics. To validate this, we evaluate the reasoning consistency (Pass@5) of the successful peer models on these divergent instances (reported as "Peer Reliability" in Table 2). We find that when Qwen2.5-Instruct-72B acts as the peer, its reasoning consistency is lower (55.4%) compared to when GPT-4.1 acts as the peer (79.1%). Consequently, the atomic questions generated by a potentially "guessing" peer may fail to pinpoint the true root cause of the error, rendering the high atomic accuracy of the student ineffective for correction. Conversely, Qwen2.5-Instruct-72B benefits from the robust, high-quality diagnosis provided by GPT-4.1, leading to a higher correction rate despite its lower atomic accuracy.

## 4.4 VERSATILITY ACROSS DIFFERENT MODELS

To validate that DGRC is a model-agnostic framework, we evaluate its performance across various teacher-student configurations on the medical domain benchmarks. As presented in Table 3, our results demonstrate significant performance improvements in all tested scenarios and reveal two key trends regarding the framework's effectiveness.

First, the curriculum's quality is positively correlated with the teacher's capability. A stronger teacher model, acting as a more proficient diagnostician, synthesizes more insightful atomic ques-

Table 3: Versatility of DGRC across different teacher and student model configurations on the medical domain. The table reports the average accuracy (%) on three medical benchmarks (Avg-M) and the relative improvement (%) over the corresponding unlabeled baseline.

| Teacher Model | Method | Qwen2.5-Instruct-1.5B | | Qwen2.5-Instruct-3B | | Qwen2.5-Instruct-7B | |
|---|---|---|---|---|---|---|---|
| | | Avg-M | Increase | Avg-M | Increase | Avg-M | Increase |
| **Qwen2.5-Instruct-32B** | Baseline w/o label | 54.3 | - | 61.5 | - | 68.7 | - |
| | DGRC (Ours) | **56.9** | (+4.9) | **63.2** | (+2.8) | **69.6** | (+1.3) |
| **Qwen2.5-Instruct-72B** | Baseline w/o label | 56.1 | - | 62.5 | - | 69.6 | - |
| | DGRC (Ours) | **59.2** | (+5.5) | **65.1** | (+4.2) | **71.1** | (+2.2) |
| **GPT-4.1 (25-04-14)** | Baseline w/o label | 54.1 | - | 62.8 | - | 73.7 | - |
| | DGRC (Ours) | **58.3** | (+7.8) | **67.1** | (+6.8) | **77.1** | (+4.6) |

Table 4: Performance of DGRC in a self-teaching configuration on the medical domain (Avg Acc. %). **Format Compliance** measures the percentage of diagnostic outputs that successfully adhered to the strict JSON format and logical constraints required for curriculum generation. The value in parentheses represents the relative improvement (%) over the zero-shot baseline. Models marked with "Failed" were unable to execute the pipeline.

| Model | Zero-shot | Format Compliance | Self-teaching | Increase |
|---|---|---|---|---|
| Qwen2.5-Instruct-1.5B | 44.0 | 12.4% | Failed | - |
| Qwen2.5-Instruct-3B | 47.9 | 34.5% | Failed | - |
| Qwen2.5-Instruct-7B | 67.3 | 78.2% | 67.9 | (+0.9) |
| Qwen2.5-Instruct-32B | 76.5 | 94.8% | 77.9 | (+1.8) |

Table 5: Ablation study on the medical domain for the Qwen2.5-Instruct-1.5B student model, with all curricula generated by the GPT-4.1 teacher.

| Training Configuration | MedMCQA | MedQA | MMLU-Med | Average |
|---|---|---|---|---|
| (1) Student (Zero-shot) | 37.1 | 40.1 | 54.8 | 44.0 |
| (2) + Atomic Curriculum (Only) | 45.3 | 42.3 | 61.6 | 49.7 |
| (3) + Verified CoT Curriculum (Only) | 46.7 | 48.3 | 68.2 | 54.4 |
| (4) DGRC (Ours) | **51.7** | **50.8** | **72.2** | **58.3** |

tions, which in turn yields a higher relative improvement for the student. Second, we observe an inverse relationship between a student's initial capability and the relative performance gain it receives. The improvement from DGRC is most pronounced for the least capable models, suggesting that weaker students, who possess more significant foundational knowledge gaps, benefit disproportionately from the targeted lessons in the atomic curriculum.

## 4.5 Self-teaching Ability

To further probe the versatility and boundaries of our framework, we investigate DGRC's capacity for self-teaching, where a model bootstraps its own reasoning capabilities by learning from its internal inconsistencies. We evaluate this on the medical domain using four models of different scales: Qwen2.5-Instruct-1.5B, -3B, -7B, and -32B. In this setup, each model diagnoses divergences among its own outputs to generate a curriculum, which is then used to fine-tune the model itself.

As shown in Table 4, DGRC enables models to improve through a self-teaching loop, but reveals a critical capability threshold. The more capable 32B model demonstrates a notable 1.8% relative improvement over its zero-shot baseline, whereas the 7B model shows a marginal gain of 0.9%. Crucially, for the smaller 1.5B and 3B models, the DGRC-style self-teaching process is failed. As evidenced by the low scores in the Format Compliance column (12.4% and 34.5%), these models struggle significantly with the instruction-following capability required to strictly adhere to the complex output constraints (e.g., JSON structure, logical isolation) for generating valid atomic Q&As. Although they produce some valid outputs, the quantity and quality are insufficient to form a robust curriculum.

This finding quantitatively characterizes the minimum capability threshold for DGRC: to act as an effective diagnostician, a model must meet a baseline of instruction-following proficiency. A

Table 6: Performance gains from an additional GRPO phase on the 7B student model on the medical domain, with all curricula generated by the GPT-4.1 teacher. The final column shows the relative improvement (%) over the SFT-only model.

| Method | MedMCQA | MedQA | MMLU-Med | Average | Increase |
|---|---|---|---|---|---|
| DGRC (SFT-only) | 67.5 | 72.8 | 91.0 | 77.1 | - |
| DGRC (SFT + GRPO) | **69.8** | **75.7** | **91.8** | **79.1** | (+2.6%) |

significant jump in compliance is observed at the 7B scale (rising to 78.2%), indicating that this is the critical inflection point where the model's ability to follow the diagnostic protocol becomes reliable enough to support self-correction.

### 4.6 THE "FROM ATOMS TO CHAINS" LEARNING TRAJECTORY

To dissect the contribution of each curriculum component, we conduct a crucial ablation study comparing four training configurations, as shown in Table 5. The results reveal two key insights. First, both the atomic curriculum and the verified CoT curriculum are independently effective, each providing a substantial performance boost over the zero-shot student. More importantly, the full DGRC framework, which trains on atoms then chains, significantly outperforms other configurations, demonstrating a clear synergistic effect.

### 4.7 EXPLORING PERFORMANCE LIMITS WITH REINFORCEMENT LEARNING

To explore the upper performance limits of DGRC, we apply an additional training phase using Group Relative Policy Optimization (GRPO) (Shao et al., 2024) to our best-performing 7B SFT-adapted model, training on the high-quality instances from the verified CoT curriculum. As shown in Table 6, this GRPO training unlocks a further +2.6% relative improvement over the SFT-only model. This result confirms that our verified CoT curriculum provides a strong signal for policy optimization, highlighting DGRC's value as a versatile data engine for advanced training paradigms.

## 5 CONCLUSION

In this work, we have introduced the Divergence-Guided Reasoning Curriculum (DGRC), a novel framework for effective LLM adaptation in unlabeled settings. Our approach is grounded in the insight of cognitive asymmetry: an LLM's atomic knowledge is reliable even when its complex reasoning fails. This insight is key to resolving the pedagogical dilemma of learning from an imperfect teacher. Specifically, DGRC introduces a mechanism that leverages reasoning divergence for an impartial diagnostic process. This process yields a two-part curriculum: an atomic curriculum to rectify foundational knowledge gaps, and a verified CoT curriculum of reliable exemplars for composing complex arguments, together actualizing the robust "from atoms to chains" learning path. Through extensive experiments, we have demonstrated that DGRC is a versatile and highly effective framework. It consistently outperforms strong baselines across multiple domains and model scales, and its structured learning path has been validated through rigorous ablation studies.

## 6 ETHICS STATEMENT

This research adheres to the ICLR Code of Ethics. Our work utilizes publicly available academic benchmarks, and all data sources and models are appropriately cited. The research process was conducted with a commitment to transparency and reproducibility. Our DGRC framework is intended as a contribution to fundamental research on automated model adaptation and is not a production-ready system for real-world decision-making. Given the high-stakes nature of the medical and legal domains, any application of this technology requires rigorous human expert oversight and validation. Furthermore, while our method employs a rigorous process to ensure quality at two levels—using a multi-stage process to filter the atomic knowledge and an atomic-knowledge-based step to verify the teacher's CoT—it does not include an explicit mechanism for debiasing; therefore, the models may still reflect biases inherited from both the source datasets and the foundational models themselves.

## 7 REPRODUCIBILITY STATEMENT

We are committed to ensuring the reproducibility of our research. A comprehensive breakdown of our implementation details can be found in Appendix F. This includes all details of our method, model configurations, datasets, and hyperparameters used in our experiments. Furthermore, we have included the full source code for our entire pipeline in the supplementary materials. We believe these resources provide all necessary components for the research community to replicate our experiments and verify our findings.

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

# Appendix

## APPENDIX CONTENTS

## A  ANSWERS CONFLICT DETECTION

This section details the methodology used to determine whether a final answer from the student model conflicts with the corresponding answer from the teacher model.

Our conflict detection process first checks for the simplest formats. For multiple-choice questions, where the answer is typically a single letter (e.g., A, B, C, D), we perform a direct string comparison after extracting and normalizing the choices. If the answer is not a simple letter, we then check for numerical or mathematical content. For these problems, we follow the standard convention of enclosing the final result within a `\boxed{}` command and utilize the `math_verify` library to check for logical equivalence. This tool robustly handles various mathematical forms, correctly identifying expressions like `\frac{1}{2}` and `0.5` as consistent.

For all other cases not covered by the above rules, such as questions requiring explanatory text, logical deductions, or other free-form answers, we resort to an LLM-as-Judge approach. To maintain a high level of domain understanding, we leverage the original teacher model itself as the impartial judge. The model is presented with the original problem, the teacher's reference answer, and the student's candidate answer, and is prompted using the template below to determine if the two are semantically consistent.

---

**Prompt Template: Answer Conflict Detection**

**Role and Task Description**
You are a precise answer verifier. Your task is to determine if Answer A and Answer B are semantically consistent and provide the same final conclusion for the given Question.

**Rules**

- **Context is Key:** Your judgment must be based entirely on the context provided by the **Question**.
- **Semantic Equivalence:** Consider two answers **CONSISTENT** if they mean the same thing, even if the wording is different. This includes synonyms, paraphrasing, and different but equivalent numerical representations (e.g., "50%", "0.5").

**Strict Output Format**
Your output MUST be one of the following two words, and nothing else. Do not provide any explanation or additional text.

- If consistent, output ONLY the string: **CONSISTENT**
- If inconsistent, output ONLY the string: **INCONSISTENT**

**Question:** {Your_Question_Here}

**Answer A:** {Your_Answer_A_Here}

**Answer B:** {Your_Answer_B_Here}

**Your Output:**

---

This tiered strategy allows us to use deterministic and precise conflict detection methods for structured answer formats, while leveraging the nuanced semantic understanding of powerful language models for more complex, open-ended responses.

## B  PROMPTS FOR CURRICULUM GENERATION

The Divergence-Guided Reasoning Curriculum (DGRC) framework relies on two core prompts during the Curriculum Generation stage. These prompts are engineered to first diagnose the underlying causes of reasoning divergence and then to verify the teacher model's reasoning chains based on the insights gained. Below, we provide the detailed structure and content of each prompt.

## B.1 PROMPT FOR ATOMIC QUESTION GENERATION

This prompt instructs the teacher model ($\mathcal{M}_T$) to act as a diagnostician. It takes a problem ($p^i$), the teacher's own reasoning chain ($c_T^i$), and the student's divergent reasoning chain ($c_S^i$) as input. The objective is to analyze both reasoning processes and synthesize a set of fundamental, self-contained "atomic questions" that pinpoint the precise factual or logical discrepancies leading to the different final answers.

The structure of the prompt is as follows:

---

**Prompt Template: Atomic Question Generation**

**# Role**

Act as an expert in logical analysis. Your specialty is deconstructing and comparing complex reasoning processes to identify the fundamental points of divergence that lead to different conclusions.

**# Task**

I will provide you with an **Original Question** and two distinct **Reasoning Processes (A and B)** from two different models, which have resulted in conflicting answers. Your task is to perform a deep analysis of both reasoning chains, pinpoint the exact discrepancies between them, and formulate these points of divergence into a set of **Atomic Questions**.

**# Core Requirements for Atomic Questions**

Each atomic question you generate must strictly adhere to the following rules:

- **Atomicity and Independence:** Each question must be the smallest possible logical unit and be completely independent. A person should be able to answer any single question on its own without needing to read any others.

- **Focus on Discrepancy:** Each question must target a specific, concrete point of disagreement between the two reasoning processes (e.g., factual claims, logical steps, or underlying assumptions).

- **Self-Contained:** If background context is necessary to understand the question, concisely embed that context within the question itself.

- **Verifiability:** Each question must be phrased so that it can be answered definitively through factual verification, a clear logical judgment, or a straightforward calculation.

**# Input Format**

**## Original Question**: {Your_Original_Question_Here}
**## Reasoning Process A**: {Reasoning_Process_A_Here}
**## Reasoning Process B**: {Reasoning_Process_B_Here}

**# Strict Output Format**

Please output in the following list format. Do not output any extra symbols or thought processes.
```
["Atomic Question 1", "Atomic Question 2", ...]
```

**\*\* NOTE \*\***

1. Do not solve the original question, just output the atomic questions.

2. When outputting atomic questions, strictly follow the specified format.

---

## B.2 PROMPT FOR VERIFIED COT CURATION

This prompt is used to filter the teacher's original candidate reasoning chains ($\mathcal{O}_T^i$). It leverages the set of high-confidence atomic question-answer pairs ($\mathcal{A}_i$) generated in the previous step as a ground-truth reference. The prompt instructs the teacher model to evaluate each of its own reasoning chains

$(c_{T,k}^i)$ for consistency against every piece of atomic knowledge in $\mathcal{A}_i$. Only chains that exhibit no contradictions are considered verified.

The structure of the prompt is as follows:

---

**Prompt Template: Verified CoT Curation**

**# Role**

You are a meticulous Factual Consistency Verifier. Your task is to act as an impartial judge, determining if a given Reasoning Chain contains any statements that contradict a provided set of established Atomic Facts.

**# Task**

I will provide you with an **Original Question**, a **Reasoning Chain** to be evaluated, and a list of **Atomic Facts** (in Q&A format) that are to be considered the absolute ground truth. Your sole task is to check for contradictions.

**# Core Rules for Verification**

- **Atomic Facts are Ground Truth:** The provided Atomic Q&A pairs are the definitive source of truth for this task. The Reasoning Chain must be evaluated strictly against them.

- **Identify Contradictions:** A contradiction exists if any part of the Reasoning Chain makes a claim that is logically or factually inconsistent with any of the Atomic Facts. This includes direct factual errors, flawed logical steps, or incorrect calculations that violate the principles established in the Atomic Facts.

- **Implicit vs. Explicit:** The contradiction can be explicit (e.g., Reasoning says "total is 30%", Fact says "total is less than 30%") or implicit (e.g., a calculation in the Reasoning violates a principle explained in a Fact).

**# Input Format**

**## Original Question:** {Your_Original_Question_Here}
**## Reasoning Chain to Verify:** {Reasoning_Chain_to_Verify_Here}
**## Atomic Facts:** {List_of_Atomic_Q&As}

**# Strict Output Format**

Your output MUST be one of the following two words, and nothing else. Do not provide any explanation.

- If the Reasoning Chain is fully consistent with ALL Atomic Facts, output ONLY the string:
  **CONSISTENT**

- If the Reasoning Chain contradicts ANY of the Atomic Facts, output ONLY the string:
  **INCONSISTENT**

**Your Output:**

---

## C  FILTERING PROCESS OF ATOMIC Q&A PAIRS

To ensure the quality, relevance, and diversity of the generated atomic knowledge, all raw question-answer pairs synthesized by the teacher model undergo a rigorous three-stage filtering process before being aggregated into the final atomic curriculum. This process is designed to discard low-quality, trivial, or redundant data.

**Instruction Following Difficulty (IFD) Filtering.** The first step filters out questions that are either too easy for the student model or show signs of misalignment. We use the Instruction Following Difficulty (IFD) (Li et al., 2023) as a proxy for this, calculated using the student model $\mathcal{M}_S$ itself. The IFD score is defined as the ratio between two intermediate scores: the conditioned answer score and the direct answer score.

For a given atomic Q&A pair $(q_{\text{atomic}}, a_{\text{atomic}})$, we define the conditioned answer score ($s_{\text{cond}}$) as the negative log-likelihood of the answer given the question. This measures how well the model can generate the answer with the instruction's context.

$$s_{\text{cond}} = -\log P_{\mathcal{M}_S}(a_{\text{atomic}}|q_{\text{atomic}}) \tag{3}$$

Then, we define the direct answer score ($s_{\text{direct}}$) as the negative log-likelihood of the answer without the question context. This gauges the inherent difficulty of generating the answer string itself.

$$s_{\text{direct}} = -\log P_{\mathcal{M}_S}(a_{\text{atomic}}) \tag{4}$$

The final IFD score is the ratio of these two values:

$$\text{IFD}(q_{\text{atomic}}, a_{\text{atomic}}) = \frac{s_{\text{cond}}}{s_{\text{direct}}} \tag{5}$$

The intuition behind this ratio is that it normalizes the difficulty of following an instruction $s_{\text{cond}}$ by the intrinsic complexity of producing the answer string itself $s_{\text{direct}}$. A low IFD score suggests that the student model can already answer the question with high confidence, indicating limited learning potential because the sample is too simple. Conversely, a high IFD score (specifically, greater than 1) suggests a misalignment between the question and answer, where the instruction provides no useful context or is even counterproductive. We therefore discard all Q&A pairs whose IFD score falls outside a predefined range $[\tau_{\text{low}}, \tau_{\text{high}}]$.

**Similarity Filtering.** Following IFD filtering, we apply similarity filtering to reduce redundancy among the atomic Q&A pairs generated from a single source problem. The core of this process is based on the cosine similarity of sentence embeddings.

For each atomic question-answer pair, we concatenate its question $q_{\text{atomic}}$ and answer $a_{\text{atomic}}$ into a single text sequence. We then use the lightweight sentence-embedding model, all-MiniLM-L6-v2 (Wang et al., 2020), to compute a single vector representation $\mathbf{v}$ for the pair. The cosine similarity between any two pairs is then calculated using the following formula:

$$\text{sim}((q_{i,\text{atomic}}, a_{i,\text{atomic}}), (q_{j,\text{atomic}}, a_{j,\text{atomic}})) = \frac{\mathbf{v}_i \cdot \mathbf{v}_j}{|\mathbf{v}_i||\mathbf{v}_j|} \tag{6}$$

where $\mathbf{v}_i$ and $\mathbf{v}_j$ are the respective embedding vectors. This calculation is the basis for both filtering stages that follow.

Within the group of Q&A pairs generated from one source problem, we conduct an all-to-all similarity comparison. If the similarity score between any two pairs exceeds a threshold $\tau_{\text{sim}}$, we apply a deterministic filtering strategy to discard the pair with the less favorable IFD score (retaining the one whose score is closer to 1).

**LLM-based Multi-dimensional Evaluation.** The final filtering step is a qualitative assessment performed by the Teacher model ($\mathcal{M}_T$). Only the pairs that have passed both IFD and similarity filters are subjected to this step. Each Q&A pair is evaluated against seven criteria: clarity, completeness, structure, credibility, knowledge richness, logicality, and instruction following ability. The total multi-dimensional score $S_{LLM}$ is calculated as the sum of these individual scores:

$$S_{LLM} = \sum_{c \in C} s_c \tag{7}$$

where $C$ represents the set of all seven evaluation criteria and $s_c \in \{0, 1, 2\}$ is the score for an individual criterion. Only atomic Q&A pairs with a total score $S_{LLM}$ above a threshold $\tau_{LLM}$ are retained for the final atomic curriculum. The detailed prompt used for this evaluation is provided below.

---

**Prompt Template: Multi-dimensional Q&A Evaluation**

**# Role**

---

You are an expert evaluator. Your task is to provide rigorous, objective quality scores for a batch of given atomic question-answer (Q&A) pairs.

# Scoring Dimensions and Rubric

For each Q&A pair, you will provide a score from 0 to 2 for each of the following dimensions.

1. **Clarity (0-2):** How clear and unambiguous is the Q&A?
   - **2:** Perfectly clear, precise, and easy to understand.
   - **1:** Mostly clear, but with minor ambiguity or awkward phrasing.
   - **0:** Vague, confusing, or poorly worded.

2. **Completeness (0-2):** Does the answer fully address the question?
   - **2:** The answer is comprehensive and fully addresses the question.
   - **1:** The answer addresses the main point but misses some nuances.
   - **0:** The answer is incomplete or fails to address the question.

3. **Structure (0-2):** Is the answer well-structured?
   - **2:** The answer is well-organized and logically structured.
   - **1:** The structure is acceptable but could be improved.
   - **0:** The answer is unstructured or chaotic.

4. **Credibility (0-2):** Is the answer factually correct and free of hallucinations?
   - **2:** The answer is factually accurate and fully credible.
   - **1:** The answer contains minor inaccuracies but is mostly correct.
   - **0:** The answer is factually incorrect or contains clear hallucinations.

5. **Knowledge Richness (0-2):** Does the answer provide sufficient, self-contained knowledge?
   - **2:** The answer provides all necessary context and is richly informative.
   - **1:** The answer is somewhat brief or lacking context.
   - **0:** The answer is too simplistic and lacks useful information.

6. **Logicality (0-2):** Is the answer logically sound?
   - **2:** The answer's internal logic is perfectly sound and coherent.
   - **1:** The answer is mostly logical but contains minor flaws.
   - **0:** The answer is illogical or contains significant reasoning errors.

7. **Instruction Following (0-2):** Does the answer adhere to all instructions and constraints?
   - **2:** The answer perfectly follows all explicit and implicit instructions.
   - **1:** The answer mostly follows instructions but has minor deviations.
   - **0:** The answer fails to follow key instructions or constraints.

# Batch Input Format

You will receive a numbered list of Q&A pairs to evaluate.
```
## Q&A Pair 1:
Question:  Question_1_Here
Answer:  Answer_1_Here
## Q&A Pair 2:
Question:  Question_2_Here
Answer:  Answer_2_Here
```
... and so on.

# Strict Output Format

You must provide your evaluation as a JSON array, where each element is a dictionary of scores. The order of the JSON objects in the array MUST correspond to the order of the Q&A pairs in the input. Your output must be ONLY the JSON array and nothing else.

```
[
{
// Scores for Q&A Pair 1
"Clarity": <0, 1, or 2>,
"Completeness": <0, 1, or 2>,
"Structure": <0, 1, or 2>,
"Credibility": <0, 1, or 2>,
"Knowledge Richness": <0, 1, or 2>,
"Logicality": <0, 1, or 2>,
"Instruction Following": <0, 1, or 2>
},
{
// Scores for Q&A Pair 2
"Clarity": <0, 1, or 2>,
"Completeness": <0, 1, or 2>,
"Structure": <0, 1, or 2>,
"Credibility": <0, 1, or 2>,
"Knowledge Richness": <0, 1, or 2>,
"Logicality": <0, 1, or 2>,
"Instruction Following": <0, 1, or 2>
}
]
```

# D  PROMPT FOR AUTOMATED ATOMIC KNOWLEDGE EVALUATION

To validate the quality of the generated atomic curriculum on a larger scale, we employ Gemini-2.5-Pro as an automated judge. We randomly sample 2,000 atomic question-answer pairs from the generated curriculum. Since ground-truth answers for these synthesized atomic questions do not exist, the model was instructed to evaluate the correctness of the answer solely based on its own expert internal knowledge. The specific prompt template used is provided below.

---

**Prompt Template: Automated Atomic Knowledge Evaluation**

# Role

You are an expert domain knowledge evaluator. Your task is to verify the correctness of an "Atomic Answer" generated for a specific "Atomic Question".

# Input Data

## Atomic Question: {atomic_question}
## Atomic Answer: {atomic_answer}

# Evaluation Rules

Please evaluate the **Atomic Answer** based on the following strict rules. You must rely on your own expert knowledge to determine validity.

- **Factual Accuracy:** The answer must be factually correct according to established consensus in the domain (e.g., medical or legal standards). It must not contain hallucinations or false information.

- **Logical Soundness:** The reasoning presented in the answer must be logically valid and coherent.

- **Responsiveness:** The answer must directly and precisely address the specific question asked, without being vague or evasive.

# Judgment Process

---

1. Read the Atomic Question and Answer carefully.

2. Verify the factual claims and reasoning against your internal expert knowledge base.

3. Determine if the answer is a correct and valid response to the question.

4. Provide a final binary verdict ("Valid" or "Invalid").

# Strict Output Format

Output a valid JSON object with the following fields:

```
{
  "reasoning": "A brief explanation of why the answer is correct or
      incorrect.",
  "verdict": "Valid" OR "Invalid"
}
```

## E    PROMPT FOR CHECK AND REWRITE THE REASONING CHAINS

Below is the prompt template used to instruct a model to act as a fact-checker and corrector in Section 4.3. The model is provided with an initial question, a corresponding reasoning, and a final prediction. Its task is to verify the reasoning and prediction against a set of provided atomic knowledge (contextual question-answer pairs) and, if necessary, correct any factual errors found.

**Prompt for Fact-Checking and Reasoning Correction**

# Task   You will be provided with a question, a model's prediction and its corresponding Chain of Thought (CoT) reasoning, along with a set of contextual question-answer pairs. Your task is to act as a fact-checker and corrector. Carefully analyze the model's CoT and its final prediction. Compare the information presented in the reasoning and prediction against the provided "Context Questions" and "Context Answers" to identify any factual inaccuracies.

# Instructions
**Review**
Scrutinize the Original CoT and Original Prediction for any statements that contradict the facts established in the Context.
**Decision**
If you find no factual errors, your output must be **ONLY** "<No>".
If you identify any factual errors, you must correct them. Your "Corrected_CoT" should begin by following the "Original CoT" up to the point of the first factual error. At that point, you must replace the incorrect statement with the accurate information from the atomic knowledge context and continue the reasoning process from there. The entire corrected reasoning process must not contradict the atomic context. Provide a "Corrected_Prediction" based on this new, accurate reasoning.

# Input Format
[Question Start] Initial question posed to the model [Question End]
[Original CoT Start] Model's step-by-step reasoning [Original CoT End]
[Original Prediction Start] Model's final answer [Original Prediction End]
[Context Question 1 Start] First contextual question [Context Question 1 End]
[Context Answer 1 Start] First contextual answer [Context Answer 1 End]
[Context Question 2 Start] Second contextual question [Context Question 2 End]

```
[Context Answer 2 Start] Second contextual answer [Context Answer 2
End] ... and so on
```

**# Output Format**
If there are no factual errors, **ONLY** output "`<No>`". Do not output any thought process or other characters.
If factual errors are found, you must output a dictionary containing the corrections. The output must be **ONLY** the dictionary and nothing else. The dictionary must be in the format as:

```
{
  "Corrected_CoT": "The revised, factually accurate step-by-step
    reasoning based on the context.",
  "Corrected_Prediction": "The new, correct final answer based on
    the corrected reasoning."
}
```

**# The data you need to process as follows:** {prompt}

## F  IMPLEMENTATION DETAILS

### F.1  DIVERGENCE DETECTION STAGE

In the divergence detection stage, we identify reasoning gaps between a teacher model and a student model using 182,822 medical questions from the MedMCQA (Pal et al., 2022) training set and 42,509 legal questions from CaseHOLD (Zheng et al., 2021) training set. For each problem, the teacher generates a single response ($K = 1$), while the student produces eight diverse responses ($J = 8$). The teacher's response is paired with each of the student's eight responses to diagnose divergences.

### F.2  CURRICULUM GENERATION STAGE

**Atomic Curriculum Generation.**   In this stage, we generate high-quality answers for the atomic questions. To enforce a consistent Chain-of-Thought format, each atomic question is appended with the instruction "`Please think carefully and then give the answer. The output format is as follows: <think> your thinking process </think><answer> your answer </answer>`" before being sent to the teacher model. The resulting atomic question-answer pairs then undergo a rigorous multi-stage filtering process as detailed in Appendix C. First, we filter based on Instruction Following Difficulty (IFD), retaining only pairs where the IFD score falls within the range of $[\tau_{\text{low}} = 0.35, \tau_{\text{high}} = 1.0]$. Next, a similarity filter is applied: for all atomic pairs from a single source question, we compute their cosine similarity, and if the score is 0.85 or higher ($\tau_{\text{sim}} = 0.85$), we discard the pair with the less favorable IFD score. Finally, the remaining pairs are subjected to a quality filter based on the teacher model's multi-dimensional evaluation; any pair with a total score below a threshold of $\tau_{LLM} = 13$ is discarded.

**Verified CoT Curriculum Generation.**   To ensure the high quality of the teacher's response to the original unlabeled question, we introduce a verification step using the generated atomic knowledge. For each problem, we gather all corresponding atomic question-answer pairs that have passed the filtering stages. From these pairs, we extract only the question and final answer to form a concise knowledge context, omitting any intermediate reasoning. This curated context is then provided to the teacher model, which re-evaluates its original CoT response for logical consistency against the provided facts. Only responses that demonstrate no logical conflicts with the atomic knowledge are retained. If multiple teacher responses for a single question pass this verification (a scenario possible when the teacher sampling count $K > 1$), we randomly select one for inclusion. This final collection of responses constitutes the verified CoT curriculum for the student adaptation stage.

### F.3  STUDENT ADAPTATION STAGE

The high-quality curricula generated in the previous stage are designed for adapting the student model, and can be used with standard Supervised Fine-Tuning (SFT) or advanced preference optimization methods like GRPO. For our main experiments, we employ a two-stage SFT process. First, the student model undergoes full-parameter fine-tuning on the atomic curriculum for 3 epochs with a learning rate of 1e-5. Subsequently, we continue full-parameter training on the verified CoT curriculum for another 3 epochs, using the same learning rate. To prevent performance degradation on simpler problems, this training stage is augmented with "no-divergence" data, where the teacher's single response was identical to all eight student responses. Detailed statistics on the data volumes for each curriculum are provided in Appendix G. For the GRPO experiments detailed in Section 4.7 and Appendix H, we adopt a different approach. The student model is initially supervised fine-tuned on the atomic curriculum for 3 epochs (full-parameter, 1e-5 learning rate), followed by a warm-up SFT phase on the verified CoT curriculum for 1 epoch with the same settings. Finally, the model is trained using GRPO for 900 steps. This phase utilizes a challenging subset of the verified CoT data from the medical domain, which is constructed using label-balanced sampling based on the final answer within the teacher model's CoT response (as opposed to the ground truth label) to include 15,000 examples for each answer option. The GRPO training proceeds with a learning rate of 1e-6 and a gradient accumulation strategy to achieve an effective batch size of 64. The SFT training is conducted using the LLaMA Factory framework (Zheng et al., 2024), while GRPO training is performed with the VERL framework (Sheng et al., 2025). All models are trained with bfloat16 precision. The setup is hardware-agnostic, and the training can be replicated on any system supporting these frameworks and bfloat16 precision. To align with our goal of domain adaptation, the training processes for the medical and legal domains are conducted independently, with each model being trained exclusively on its corresponding domain-specific curriculum.

### F.4  INFERENCE AND EVALUATION

**Inference Details.**    Throughout all stages of curriculum generation and evaluation, we adhere to a consistent set of inference parameters. For closed-source models, we utilize their official APIs for all interactions. For open-source models, we leverage the vLLM framework (Kwon et al., 2023) for efficient inference and employ a standard sampling configuration with a temperature of 0.6, top-k of 30, and top-p of 0.95. All inference is conducted with bfloat16 precision, and the maximum context length is set to 4096 tokens.

**Benchmark Evaluation.**    To facilitate answer extraction during benchmark evaluation, we uniformly append the following instruction to each question: "`Please think step-by-step and then give the final result.  The output format is as follows: <think> your thinking process </think><answer> The final answer should be a single capital letter.  </answer>`" To ensure the reliability and stability of our results on public benchmarks, we mitigate potential scoring fluctuations by running each evaluation 10 times. The final reported performance is the average of these 10 runs.

## G  DETAILED CURRICULUM STATISTICS

This section provides a detailed breakdown of the curricula generated for our main experiments, the results of which are presented in Table 1 and Table 3. All statistics reported here correspond to our primary experimental configuration, using a teacher sampling count of $K = 1$ and a student sampling count of $J = 8$.

The data generation pipeline, summarized in Figure 2, begins with a set of unlabeled problems. For each problem, a problem is categorized as a divergent problem (# Div. Problems) if at least one of the eight student responses conflicts with the teacher's single response. Problems where all eight student responses are consistent with the teacher's are considered "no-divergence" problems. The # Div. Pairs column counts the total number of individual student-teacher response pairs that exhibit a conflict across all divergent problems.

From these divergent pairs, we generate an initial set of raw atomic questions (# Raw Atomic Q&A). This set is then refined through our multi-stage filtering process to produce the final filtered atomic

Table 7: Detailed statistics of the curriculum generation pipeline across different domains and models.

| Domain | Teacher Model | Student Model | # Unlabeled Problems | # Div. Problems | # Div. Pairs | # Raw Atomic Q&A | # Filtered Atomic Q&A | # Verified CoTs | # Total CoTs | Avg. Tokens (Atomic) | Avg. Tokens (CoT) |
|---|---|---|---|---|---|---|---|---|---|---|---|
| Medical | GPT-4.1 | Qwen2.5-Instruct-7B | 182,822 | 106,661 | 485,719 | 1,892,496 | 413,886 | 88,698 | 164,859 | 171.7 | 188.6 |
| | | Qwen2.5-Instruct-3B | 182,822 | 155,357 | 835,136 | 3,674,964 | 701,256 | 121,441 | 148,906 | 166.3 | 190.6 |
| | | Qwen2.5-Instruct-1.5B | 182,822 | 120,437 | 810,860 | 3,757,310 | 560,943 | 103,159 | 165,544 | 170.5 | 189.5 |
| | Qwen2.5-Instruct-72B | Qwen2.5-Instruct-7B | 182,822 | 104,623 | 444,983 | 3,074,152 | 721,970 | 72,856 | 151,055 | 169.0 | 250.5 |
| | | Qwen2.5-Instruct-3B | 182,822 | 154,782 | 820,373 | 4,701,122 | 868,242 | 109,940 | 137,980 | 172.0 | 256.3 |
| | | Qwen2.5-Instruct-1.5B | 182,822 | 118,554 | 793,918 | 4,712,338 | 709,901 | 80,080 | 144,348 | 169.9 | 261.2 |
| | Qwen2.5-Instruct-32B | Qwen2.5-Instruct-7B | 182,822 | 103,051 | 437,653 | 1,532,535 | 330,136 | 60,641 | 140,412 | 133.3 | 256.1 |
| | | Qwen2.5-Instruct-3B | 182,822 | 154,631 | 816,849 | 2,999,135 | 525,273 | 92,813 | 121,004 | 131.8 | 266.3 |
| | | Qwen2.5-Instruct-1.5B | 182,822 | 117,527 | 786,991 | 2,902,090 | 394,323 | 71,234 | 136,529 | 135.5 | 254.4 |
| Legal | GPT-4.1 | Qwen2.5-Instruct-7B | 42,509 | 23,631 | 104,082 | 405,648 | 86,335 | 16,058 | 34,936 | 178.1 | 367.8 |
| | | Qwen2.5-Instruct-3B | 42,509 | 36,296 | 201,928 | 793,505 | 133,715 | 24,418 | 30,631 | 177.8 | 354.6 |
| | | Qwen2.5-Instruct-1.5B | 42,509 | 28,234 | 182,173 | 741,298 | 113,090 | 19,378 | 33,653 | 181.6 | 357.9 |

Table 8: Performance comparison to evaluate the effectiveness of the CoT verification step on the medical domain. The student model is a 7B parameter model, and the teacher model is GPT-4.1. All models are first fine-tuned on the same atomic curriculum. Scores are reported as accuracy (%) on three benchmarks and their average.

| Training Method | CoT Curriculum Type | MedMCQA | MedQA | MMLU-M | Average |
|---|---|---|---|---|---|
| SFT | Original CoT (w/o Verification) | 66.1 | 70.1 | 89.3 | 75.2 |
| | Verified CoT (Ours) | **67.5** | **72.8** | **90.9** | **77.1** |
| GRPO | Original CoT (w/o Verification) | 67.8 | 74.3 | 91.6 | 77.9 |
| | Verified CoT (Ours) | **69.8** | **75.7** | **91.8** | **79.1** |

curriculum (# Filtered Atomic Q&A). Concurrently, the teacher's original responses for the divergent problems undergo verification, resulting in a set of verified CoTs (# Verified CoTs). The total CoT we used (# Total CoTs) is then formed by combining these verified CoTs with the single CoT from each no-divergence problem. Finally, we report the average token lengths for the two curricula.

## H EFFECTIVENESS OF CoT VERIFICATION

To isolate and quantify the benefit of our Chain-of-Thought (CoT) verification step, we conduct a targeted ablation study. We compare the performance of student models trained on two different CoT curricula, subsequent to an initial training phase on the same atomic curriculum. The baseline model is trained using the teacher's original CoT responses, which have not undergone our verification process. In contrast, our proposed model is trained using the verified CoT curriculum, where responses with logical inconsistencies have been filtered out. We evaluate this comparison under two distinct training paradigms to demonstrate the robustness of our approach: SFT and GRPO.

The results, presented in Table 8, demonstrate that the CoT verification step provides a performance boost. In both the SFT and GRPO settings, the student model trained on the verified curriculum consistently outperforms the baseline model trained on the original unverified CoT. This confirms that eliminating flawed reasoning chains from the teacher's responses is crucial for generating a higher-quality training signal for the student model.

## I ORTHOGONALITY AND SYNERGY WITH REINFORCEMENT LEARNING

In Table 1, we compared DGRC against a Reinforcement Learning from GPT-4.1 Feedback (RLAIF) baseline implemented via Group Relative Policy Optimization (GRPO). While this comparison establishes DGRC's competitiveness, it is crucial to clarify that DGRC and RLAIF are not mutually exclusive alternatives but rather orthogonal and complementary components of the LLM alignment pipeline.

**Conceptual Orthogonality.** Fundamentally, DGRC and RLAIF operate at distinct yet complementary stages of the alignment pipeline. DGRC functions primarily as a data synthesis engine on the input side, leveraging the cognitive asymmetry to synthesize high-quality curricula from unlabeled queries. Its core contribution lies in filtering the noise and hallucinations inherent in self-generated data to produce reliable supervision. In contrast, RLAIF represents a training paradigm on the optimization side, which presupposes the existence of a reward signal and focuses on optimizing the model's policy to maximize that reward. Thus, rather than competing, the two approaches form a cohesive system where DGRC provides the high-quality reasoning paths, while RLAIF utilizes these paths for effective policy optimization.

**DGRC as a Superior Foundation for Low-Resource Models.** Our results in Table 1 demonstrate that DGRC consistently outperforms the RLAIF baseline in lower-parameter regimes (e.g., +2.6% over RLAIF for the 1.5B model). This finding highlights a critical insight: For smaller models with limited reasoning capabilities, explicit instruction is more effective than implicit feedback. RLAIF relies on the model to self-explore and discover correct reasoning paths based on sparse reward signals, a task that is often too challenging for smaller models with limited search space. In contrast,

Table 9: Ablation study on teacher (K) and student (J) sampling counts. The student model is Qwen2.5-Instruct-1.5B fine-tuned on a curriculum generated by GPT-4.1. Values represent the **average accuracy (%)** across three medical benchmarks: MedMCQA, MedQA, and MMLU-Medical.

| K \ J | 1 | 2 | 4 | 8 |
|---|---|---|---|---|
| 1 | 53.5 | 55.0 | 56.8 | 58.3 |
| 2 | 55.8 | 55.7 | 57.2 | 58.5 |
| 3 | 56.4 | 58.1 | 58.6 | **59.2** |

DGRC decomposes complex problems into atomic steps, providing dense and explicit supervision that acts as a scaffold, making it significantly easier for these models to acquire domain knowledge.

**Synergistic Potential.** The true potential lies in combining these two approaches, as evidenced by our ablation studies across Section 4.7 and Appendix H. First, DGRC serves as a powerful "warm-up" for RL: applying GRPO on top of the DGRC-adapted model yields further improvements, pushing the 7B model from 77.1% to 79.1%, as illustrated in Table 6. Second, the Atomic Curriculum specifically optimizes the starting policy for RL: as shown in Table 8, fine-tuning on atomic questions before applying GRPO outperforms the direct RLAIF baseline by 1.7% (77.9% vs. 76.2% in Table 1). Finally, the quality of the training data matters: using DGRC's Verified CoT Curriculum for GRPO training provides a clear advantage over using unverified teacher responses, further driving performance from 77.9% to 79.1%, as shown in Table 8. This confirms that DGRC and RLAIF can be pipelined effectively: DGRC first establishes a robust reasoning foundation and cleans the data, creating a high-quality policy that allows subsequent RLAIF stages to focus on refining complex reasoning paths.

## J ABLATION STUDY ON SAMPLING COUNT

We conduct an ablation study to analyze the impact of the teacher sampling count ($K$) and the student sampling count ($J$). Using the GPT-4.1 teacher and Qwen2.5-Instruct-1.5B student on a 10,000-question subset from the MedMCQA training data, we experiment with $K \in \{1, 2, 3\}$ and $J \in \{1, 2, 4, 8\}$. The results are presented in Table 9, revealing two clear trends. First, increasing the student sampling count $J$ consistently improves performance. A higher $J$ increases the probability of exposing latent reasoning deficiencies in the student model, which in turn allows for the generation of more targeted atomic knowledge for its correction. Second, a higher teacher sampling count $K$ also leads to better results. This is because a larger sample pool increases the likelihood of obtaining a high-quality CoT consistent with the atomic knowledge, thereby enriching the final verified CoT curriculum. The peak performance is achieved at $K = 3$ and $J = 8$, confirming the benefits of enriching the verified CoT curriculum (via $K$) while simultaneously improving the targeting of the atomic curriculum (via $J$). For our main experiments, we set $J = 8$ to maximize the discovery of student weaknesses. However, we use $K = 1$ primarily to ensure high experimental efficiency.

## K QUALITATIVE ANALYSIS

This section presents a qualitative analysis to demonstrate the effectiveness of our atomic questions. We analyze examples from the MedMCQA validation set where the student model makes errors, visualizing its internal uncertainty and aligning it with the corresponding atomic question designed to address the error.

**Entropy as a Measure of Model Uncertainty.** To visualize the student model's token-level uncertainty, we use Shannon Entropy. Entropy quantifies the uncertainty in the model's predictive distribution for the next token: a higher value signifies greater confusion, while a lower value suggests higher confidence. For a predictive distribution $P$ over the vocabulary $V$, the entropy $H$ is calculated as:

$$H(P) = - \sum_{w \in V} P(w) \log_2 P(w) \tag{8}$$

where $P(w)$ is the probability of token $w$. In our visualizations (Figure 3), higher token entropy is represented by a more intense background color, visually highlighting where the model is most uncertain.

**Sourcing Atomic Questions for Analysis.** Our qualitative analysis relies on source attribution to link each atomic question to its origin in the student's reasoning, allowing for an evaluation of its relevance and quality. We achieve this using a specialized prompt that instructs the teacher model to both formulate focused atomic questions based on the reasoning divergency and extract the precise source sentence containing the core misconception from the student's response. The prompt we used is as follows:

---

### Atomic Question Generation with Source Attribution

#### # Role and Task
Act as an expert in logical analysis specializing in deconstructing and comparing complex reasoning processes. You will be provided with an Original Question and two distinct Reasoning Processes (A and B) that have resulted in conflicting answers. Your task is to perform a comprehensive analysis of both reasoning chains, identify all significant logical discrepancies, and formulate an atomic question for each distinct point of divergence.

#### # Instructions
Your goal is to generate a complete list of atomic questions that covers every fundamental conflict between the two reasoning processes. Each question and its attributed sources must strictly adhere to the following rules:

- **Atomicity and Independence:** Each question must be the smallest possible logical unit and be completely independent of all other questions.

- **Focus on Discrepancy:** Each question must target a specific, concrete point of disagreement between the two reasoning processes.

- **Self-Contained:** If background context is necessary to understand the question, concisely embed that context within the question itself.

- **Verifiability:** Each question must be phrased in a way that it can be answered definitively through factual verification, a clear logical judgment, or a straightforward calculation.

- **Bilateral Source Attribution:** For each atomic question, you must locate and extract the corresponding critical text snippets from **both** Reasoning Process A and Reasoning Process B. These two snippets should clearly represent the direct point of conflict.

#### # Input Format
```
Original Question:  {problem}
Reasoning Process A: {cot1}
Reasoning Process B: {cot2}
```

#### # Output Format
You must strictly output a JSON array of objects and nothing else. The array should contain one object for each distinct discrepancy you identify. Each object must have three keys: `"atomic_question"`, `"source_from_A"`, and `"source_from_B"`.
Example format:

```
[
  {
    "atomic_question": "Content of the first atomic question.",
    "source_from_A": "The specific text snippet from Reasoning
        Process A that is directly related to the point of
        discrepancy.",
```

---

```
      "source_from_B": "The specific text snippet from Reasoning
          Process B that is directly related to the point of
          discrepancy."
    },
    {
      "atomic_question": "Content of the second atomic question.",
      "source_from_A": "Another text snippet from Reasoning Process A
          related to the second point of discrepancy.",
      "source_from_B": "Another text snippet from Reasoning Process B
          related to the second point of discrepancy."
    }
  ]
```

**# Final Constraints**

- Do not solve the original question; your task is only to generate the atomic questions for comparison.

- Strictly adhere to the JSON array format. Do not output any extra explanations or thought processes.

- Ensure the value for the `"atomic_question"` key fulfills all the core requirements.

- Ensure the values for the `"source_from_A"` and `"source_from_B"` keys are precise, concise, direct quotes that clearly showcase the conflict.

**Analysis.** By analyzing the qualitative examples presented in Figure 3, we derive two key insights. First, we observe that the student model's high-entropy regions are not randomly distributed but are concentrated in its reasoning about both the **incorrect answer it selected** and the **correct ground-truth answer**. This indicates more than a simple mistake; it points to a deeper conceptual confusion where the model is most uncertain precisely when it has to discriminate between critical, competing concepts. This finding suggests that high entropy is a indicator of a model's specific knowledge gaps. Second, we find that the source sentences identified by our method for generating atomic questions consistently exhibit high entropy. This demonstrates that our atomic questions are precisely targeted at the model's points of maximum uncertainty and error. This validates the efficacy of our atomic curriculum, showing it functions as a targeted intervention tool. By focusing directly on diagnosed weaknesses, the curriculum provides a foundation for an efficient and focused method of model enhancement, addressing the very concepts the model struggles with the most.

## L   DATA CONTAMINATION AND GENERALIZATION ANALYSIS

A critical concern in post-training adaptation is whether performance gains stem from genuine reasoning improvements or simply from data contamination (i.e., the model memorizing benchmarks seen during pre-training or leaked from the teacher). To address this, we analyze contamination risks from both the teacher and student perspectives.

**Teacher-Side Contamination and OOD Generalization.**   A primary risk involves the powerful teacher models (e.g., GPT-4.1) potentially having the test benchmarks in their pre-training corpus, thereby leaking ground-truth information into the generated curriculum. To rigorously assess this, our experimental design strictly enforces a cross-dataset evaluation setup, ensuring that the source training data is distinct from the evaluation benchmarks used to measure generalization.

- **Medical Domain:** The curriculum is generated exclusively from the MedMCQA training set. We then evaluate generalization on MedQA and MMLU-Medical, which serve as Out-Of-Distribution (OOD) benchmarks.

- **Legal Domain:** The curriculum is generated exclusively from CaseHOLD. We evaluate generalization on MMLU-Law, an OOD benchmark.

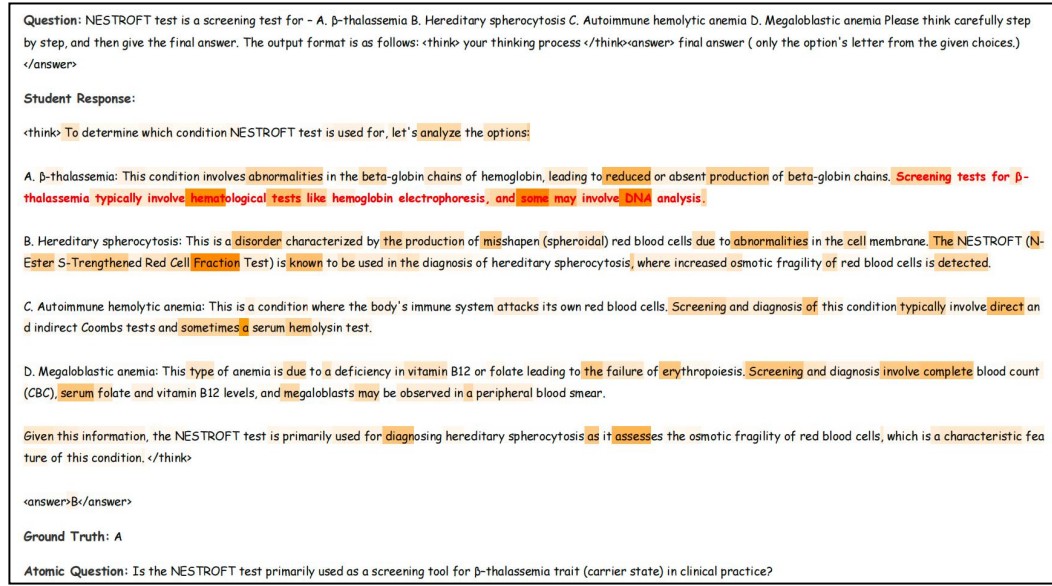

Figure 3: Qualitative examples of student model errors. Token-level entropy is visualized by color intensity. The generated atomic question targets the specific reasoning error, whose source sentence is highlighted in red, which consistently aligns with a high-entropy region.

Table 10 presents the performance on these OOD benchmarks. We observe consistent and substantial performance gains across all metrics. For instance, the 1.5B model achieves an average improvement of **+4.9%** over the baseline on OOD tasks. This strong OOD generalization effectively rules out simple teacher-side data leakage as the primary driver of performance; instead, it indicates that DGRC successfully instills transferable atomic knowledge and reasoning capabilities that generalize to unseen tasks.

**Student-Side Contamination Risks.** Since the student models (Qwen2.5 series) are open-source and trained on massive web corpora, there is an inherent risk that they may have encountered the test benchmarks during their pre-training phase. However, as detailed in Table 10, our DGRC method yields dramatic performance improvements over the original Zero-shot models (e.g., **+11.6%** for

Table 10: Analysis of Out-Of-Distribution (OOD) generalization and Student-Side contamination risks. We compare the original open-source model (Zero-shot), the distillation baseline, and our DGRC method on three OOD benchmarks. **Avg OOD** represents the mean accuracy across these unseen tasks. The consistent improvement over the Zero-shot baseline confirms that DGRC enhances reasoning beyond potential pre-training memorization.

| Model | Method | MedQA | MMLU-Med | MMLU-Law | Avg OOD | Gain vs. Zero-shot |
|-------|--------|-------|----------|----------|---------|--------------------|
| Qwen2.5-1.5B | Zero-shot (Original) | 40.1 | 54.8 | 61.2 | 52.0 | - |
| | Baseline-w/o-label | 48.3 | 65.0 | 62.8 | 58.7 | +6.7 |
| | **DGRC (Ours)** | **50.8** | **72.2** | **67.8** | **63.6** | **+11.6** |
| Qwen2.5-3B | Zero-shot (Original) | 41.6 | 63.7 | 61.2 | 55.5 | - |
| | Baseline-w/o-label | 58.4 | 73.5 | 69.4 | 67.1 | +11.6 |
| | **DGRC (Ours)** | **60.6** | **81.0** | **72.7** | **71.4** | **+15.9** |
| Qwen2.5-7B | Zero-shot (Original) | 59.8 | 86.6 | 75.2 | 73.9 | - |
| | Baseline-w/o-label | 69.4 | 87.2 | 78.5 | 78.4 | +4.5 |
| | **DGRC (Ours)** | **72.8** | **90.9** | **79.3** | **81.0** | **+7.1** |

1.5B and **+15.9%** for 3B). If the student models were merely recalling memorized answers from pre-training contamination, we would expect the Zero-shot performance to be closer to the adapted performance, or for the adaptation to yield diminishing returns. Instead, the significant leap in accuracy demonstrates that DGRC is not simply unlocking memorized data, but is actively constructing new, domain-specific reasoning pathways that the base models originally lacked. Furthermore, DGRC consistently outperforms the Baseline-w/o-label, confirming that our curriculum-based approach is superior to standard distillation in leveraging these new pathways.

# M   COMPUTATIONAL COST ANALYSIS

To rigorously quantify the resource implications of our method, we present a comparative analysis between DGRC and the standard unlabeled distillation baseline (Baseline-w/o-label). This analysis is based on the experimental statistics from the medical domain presented in Table 7, specifically focusing on the configuration where GPT-4.1 serves as the teacher and Qwen2.5-Instruct-7B as the student ($N = 182,822$ unlabeled problems). We detail the token consumption across the pipeline, accounting for our batching and hierarchical filtering strategies.

**Breakdown of Resource Consumption.**

- **Offline Generation Phase (One-time Cost):**
    - *Diagnosis & Decomposition:* For the divergent pairs (approx. 2.6 pairs per problem), the teacher generates atomic questions in a single batch. This amortizes the input context cost, making the process efficient.
    - *Batch Answering:* We instruct the teacher to answer all generated atomic questions for a problem in a single inference pass, significantly reducing the overhead compared to answering them individually.
    - *Hierarchical Filtering:* We employ a "funnel" strategy to minimize the use of the expensive LLM judge. Low-cost filters (IFD calculation by the small student model and embedding-based similarity) are applied first to discard trivial or redundant pairs. Only the high-quality candidates ($\sim 50\%$ of raw pairs) are sent to the teacher for multi-dimensional scoring, saving substantial compute.
    - *Overall:* While the teacher inference load is roughly $6\times$ that of the baseline, this cost is incurred only once during dataset creation and is negligible compared to pre-training costs.
- **Online Training Phase:** The training overhead is moderate and linear. The DGRC curriculum (Atomic + Verified CoT) contains roughly $2.5\times$ the number of tokens compared to the baseline dataset. Consequently, training takes approximately $2.5\times$ longer. However, unlike RL-based methods which require complex infrastructure and multiple models in memory, DGRC utilizes standard SFT, keeping the GPU memory footprint identical to the baseline.

**Cost-Performance Trade-off.**   For resource-constrained scenarios, DGRC offers flexible deployment options:

Table 11: Computational cost vs. performance comparison. Calculations are normalized per unlabeled problem. Token counts are estimated averages: Problem Context $\approx 200$, CoT $\approx 400$, Atomic QA Pair $\approx 100$. DGRC uses $K = 1, J = 8$. Note that the generation cost is a one-time offline investment.

| Phase | Metric | Baseline (Distillation) | DGRC (Standard, $J = 8$) |
|---|---|---|---|
| **Offline Generation** | Teacher Inference (Tokens) | $\sim 400$ (1 Response) | $\sim 2,500$ (Gen + Diag + Batch Ans + Ver) |
| | Student Inference (Tokens) | 0 | $\sim 3,200$ (8 Responses) |
| | *Primary Cost Driver* | *Single Generation* | *Generation & Diagnosis & LLM Filtering Judge* |
| | *Cost Multiplier* | $1.0\times$ | $\approx 6.25\times$ (Teacher)$+ 8\times$ (Student) |
| **Online Training** | Dataset Size (Instances) | $\sim 182k$ CoTs | $\sim 413k$ Atomic $+ \sim 88k$ Verified CoTs |
| | Total Training Tokens | $\sim 72$M | $\sim 180$M |
| | *Relative Compute* | $1.0\times$ | $\approx 2.5\times$ |
| **Outcome** | **Avg Accuracy** | 73.7% | **77.1%** |

- **DGRC-Standard** ($J = 8$)**:** Used in our main experiments to maximize performance by uncovering diverse reasoning gaps. Suitable for high-stakes domains like medicine.

- **DGRC-Lite** ($J = 2$ **or** $4$)**:** Reducing the student sampling count linearly decreases the diagnosis cost. As indicated in Appendix J, even $J = 2$ yields significant improvements, offering a balanced solution that cuts the generation cost by roughly half while retaining most of the performance gains.

## N    LIMITATION

While our DGRC framework demonstrates significant promise, we acknowledge two primary limitations that warrant further investigation.

**Dependence on Teacher Model Capability.**    The efficacy of DGRC is highly dependent on the capability gap between the teacher and student models. As shown in experiment in Section 4.4, the largest performance gains are achieved when a state-of-the-art model like GPT-4.1 serves as the teacher, and there is a clear trend where stronger teachers yield greater improvements. This reliance is further highlighted in the self-teaching setting in Section 4.5, where the more capable 32B model improves notably more than the 7B model because the weaker model's ability to act as a reliable "diagnostician" becomes a significant bottleneck. This suggests that the practical applicability of DGRC may be constrained in scenarios where access to significantly more powerful teacher models is limited or cost-prohibitive.

**Risk of Shared Blind Spots.**    A second limitation is the risk of "shared blind spots", where both the teacher and student models make the same error. Since our framework is triggered by reasoning divergence, such cases of shared error will go undetected, preventing the generation of a corrective lesson. To address this, we see significant potential in integrating DGRC with external knowledge bases (i.e., Retrieval-Augmented Generation Lewis et al. (2020)). We envision a hybrid detection mechanism: for instances where the teacher and student reach a consensus (no divergence), the system could retrieve relevant evidence from a reliable external domain corpus. If a conflict arises between the models' consensus and the retrieved knowledge, it would flag a potential shared blind spot, allowing the external evidence to serve as the ground truth for generating corrective atomic curricula. Conversely, alignment with retrieved knowledge would further validate the reliability of the consensus.

**Contextualizing the Limitations.**    It is important, however, to contextualize these limitations. Dependence on a capable teacher is an inherent characteristic of most knowledge distillation frameworks, as well as methods that synthesize data from external knowledge bases. Separately, while the challenge of shared blind spots is a common hurdle for all approaches that operate in unlabeled settings, our proposed integration with external knowledge bases offers a concrete pathway to mitigate this issue. Furthermore, while performance is maximized with a state-of-the-art proprietary model, our results demonstrate that DGRC remains highly effective when using strong open-source models as teachers. As shown in Table 3, models like Qwen2.5-Instruct-72B still yield substan-

tial improvements over the baseline, highlighting the framework's practical value and potential for broader application in various research contexts.

## O  THE USE OF LARGE LANGUAGE MODELS

The authors acknowledge the use of a large language model (LLM) as a writing assistant during the preparation of this manuscript. Its application was strictly confined to language enhancement tasks, such as improving the text's clarity, conciseness, and grammatical correctness. The core scientific contributions, including the conceptualization of the DGRC framework, the experimental design, and the analysis of the results, are the original work of the authors. All suggestions provided by the LLM were critically reviewed, edited, and approved by the authors, who take full responsibility for the final content of this paper.

