# OpenReview forum: "From Atoms to Chains: Divergence-Guided Reasoning Curriculum for Unlabeled LLM Domain Adaptation"
_ICLR.cc/2026/Conference — Submitted to ICLR 2026_

### Official Review · Reviewer_C29g · 2025-10-22

**Soundness:** 3
**Presentation:** 3
**Contribution:** 3
**Rating:** 6
**Confidence:** 3

**Summary:**

This paper introduces the Divergence-Guided Reasoning Curriculum (DGRC), an innovative framework designed to address the critical challenge of adapting Large Language Models (LLMs) to specialized domains without human-annotated data. The DGRC framework cleverly leverages divergences in reasoning paths between a "teacher" and a "student" model as a trigger for learning. Instead of simply assuming the student is wrong, DGRC treats divergence as an opportunity for a neutral, diagnostic inquiry. The teacher model analyzes both reasoning chains to generate "atomic questions" targeting the specific points of disagreement. It then re-answers these questions to create a high-confidence "atomic curriculum." Concurrently, this verified atomic knowledge serves as a factual standard to filter the teacher's original reasoning chains, yielding a "verified CoT curriculum." The student model is then trained via an "atom-to-chain" paradigm, first rectifying foundational knowledge gaps with the atomic curriculum and then mastering compositional reasoning with the verified CoT curriculum.

**Strengths:**

1. The concept of "Cognitive Asymmetry" elegantly captures a key characteristic of current LLM capabilities. Reframing model disagreement from a simple "student error signal" to a "trigger for impartial diagnosis" is a highly intelligent and novel perspective. This successfully circumvents the fundamental dilemma of learning from a "fallible teacher" in unlabeled settings.
2. Through an automated pipeline, DGRC transforms abstract model disagreements into two concrete, complementary curricula. This design adeptly solves two core problems: (1) how to precisely identify and rectify the student's knowledge gaps (via the atomic curriculum), and (2) how to ensure the teacher's supervision is reliable (via the verified CoT curriculum).
3. The paper reports significant performance gains, such as a 7.76% relative improvement for the 1.5B student model in the medical domain. Furthermore, DGRC demonstrates remarkable parameter efficiency and generalization; for instance, the DGRC-adapted 7B model outperforms its 32B baseline counterpart, highlighting its practical value for deploying smaller models in resource-constrained environments.

**Weaknesses:**

1. The DGRC pipeline is quite complex, involving multiple rounds of LLM sampling, divergence diagnosis, atomic question generation and answering, and CoT verification. This process is computationally expensive. This point is particularly salient given that a primary motivation for distillation-based methods is to create smaller, efficient models. If the curriculum generation process itself is prohibitively expensive, it risks becoming counterproductive, undermining the very goal of achieving cost-effective performance gains in the student model. It would be beneficial for the authors to provide a quantitative analysis of the computational cost (e.g., total tokens or API calls per sample), as this is crucial for assessing the method's practical viability.
2. The experimental results (Tables 3 and 4) clearly indicate that DGRC's effectiveness is positively correlated with the teacher model's capability. While the authors acknowledge this in the limitations section, it is a primary practical bottleneck. The method's efficacy might be substantially reduced in scenarios where access to powerful models like GPT-4.1 is unavailable.
3. As the framework is triggered by divergence, it cannot detect or correct errors that are common to both the teacher and student models("shared blind spots"). The authors mention this, but its potential impact might be understated. A discussion of potential mitigation strategies (such as incorporating a third-party model or an external knowledge base as an judger)would strengthen the paper.

**Questions:**

1. Can you quantify the computational cost of the DGRC pipeline in comparison to the baseline distillation methods?
2. Regarding the "shared blind spots" issue, do you see potential in integrating the DGRC framework with external knowledge bases to detect and correct such errors?
3. In the peer-correction experiment (Section 4.3), Qwen2.5-Instruct-72B achieved a higher final correction rate than GPT-4.1, despite having lower atomic knowledge accuracy. This is an interesting finding. Could you elaborate on your hypotheses for this phenomenon?

---

> ### Author Response · Authors · 2025-11-26
> **Response1 to Reviewer C29g**
>
> We sincerely thank the reviewer for the detailed and positive assessment. We appreciate your recognition of "Cognitive Asymmetry" as an elegant concept and your validation of our automated pipeline as a solution to the "fallible teacher" dilemma. We have updated our manuscript to provide the requested quantitative cost analysis and to elaborate on the peer-correction phenomenon.
>
> **1. Quantitative Analysis of Computational Cost**
>
> > **Q:** Can you quantify the computational cost of the DGRC pipeline in comparison to the baseline distillation methods?
>
> We agree that practical viability is crucial. We have added a comprehensive **Cost-Performance Analysis** in **Appendix M** and **Table 11**.
>
> * **Offline Generation (One-time Cost):** The DGRC pipeline involves higher upfront compute for the teacher model due to diagnosis and verification. Specifically, the teacher inference load is approximately **6.25x** that of the standard distillation baseline. However, we employ a "funnel" strategy (using low-cost IFD and similarity filters first) to minimize the use of the expensive LLM judge. Crucially, this is a *one-time* offline cost for dataset creation.
> * **Online Training:** The DGRC curriculum (Atomic + Verified CoT) contains roughly **2.5x** the number of tokens compared to the baseline dataset (due to the addition of atomic Q&As). Consequently, training time increases linearly by $\approx 2.5\times$. However, unlike RL-based methods that require multiple models in memory, DGRC uses standard Supervised Fine-Tuning (SFT), keeping the GPU memory footprint identical to the baseline.
> * **Trade-off:** For resource-constrained scenarios, we introduce **"DGRC-Lite"** (discussed in **Appendix M**). Reducing the student sampling count from $J=8$ to $J=2$ cuts the generation cost by nearly half while retaining the majority of the performance gains, offering a flexible balance between cost and accuracy.
>
> **2. Shared Blind Spots and External Knowledge Bases**
>
> > **Q:** Regarding the "shared blind spots" issue, do you see potential in integrating the DGRC framework with external knowledge bases...?
>
> Yes, we strongly agree with this direction. As discussed in our updated **Section N (Limitations)**, we view the integration of **Retrieval-Augmented Generation (RAG)** as the definitive solution for shared blind spots.
>
> * **Proposed Mechanism:** We outline a "Hybrid Detection" strategy: when the teacher and student reach a consensus (no divergence), the system could retrieve relevant evidence from a trusted external corpus. A conflict between the model consensus and the retrieved evidence would act as a new trigger, flagging a "shared blind spot" and allowing the external evidence to serve as the ground truth for generating corrective curricula.
> * **Current Contribution:** While this is a promising future direction, our current work focuses on maximizing the utility of the internal knowledge within the model pair. By resolving disagreements, we significantly improve reliability even before external resources are introduced.
>
> **3. Hypothesis on Correction Rate Discrepancy**
>
> > **Q:** In the peer-correction experiment ... Qwen2.5-Instruct-72B achieved a higher final correction rate than GPT-4.1, despite having lower atomic knowledge accuracy... Could you elaborate?
>
> This is a interesting finding that we have investigated further in **Section 4.3**. We hypothesize that this discrepancy is driven by the **reliability of the peer's reasoning** used for diagnosis.
>
> * **The Mechanism:** In our setup, correction is triggered by divergence. When a model fails (e.g., GPT-4.1), the curriculum is derived from the reasoning of its successful peer (e.g., Qwen-2.5-72B).
> * **Peer Reliability:** A correct final answer does not guarantee correct reasoning—the peer might have "guessed" correctly via flawed heuristics on hard problems. To validate this, we added a **"Peer Reliability"** metric (Pass@5 consistency) to **Table 2**.
> * **The Explanation:** We found that when Qwen-2.5-72B acted as the peer, its reasoning consistency was significantly lower (**55.4%**) compared to when GPT-4.1 acted as the peer (**79.1%**).
> * **Conclusion:** When GPT-4.1 fails, it generates atomic questions based on the "diff" with the "guessing" peer (Qwen-2.5-72B). Since Qwen's reasoning path was likely flawed (despite the correct answer), the resulting atomic questions, while factually answerable by GPT-4.1 (95.0% accuracy), target non-essential differences, failing to fix the root error. Conversely, when Qwen-2.5-72B fails, it benefits from diagnosing against the high-quality reasoning of GPT-4.1, leading to a higher correction rate (34.6%) despite its lower atomic accuracy.

---

> > ### Comment · Reviewer_C29g · 2025-11-27
> >
> > I appreciate the authors' effort in addressing my comments, particularly the quantitative analysis of computational costs and the clarification on the peer-correction mechanism. The response has resolved my main questions. I believe the proposed DGRC framework is a valuable contribution to domain adaptation, and I will keep my original score.

---

### Official Review · Reviewer_4rkB · 2025-10-28

**Soundness:** 2
**Presentation:** 2
**Contribution:** 2
**Rating:** 4
**Confidence:** 4

**Summary:**

This paper studies unlabeled LLM domain adaptation for student model based on the divergence-guided reasoning curriculum. The proposed method DGRC first decomposes complex query into atom queries which are derived from the divergence between teacher and student models. Then DGRC steps into the curriculum generation stage and student adaptation stage. Extensive experiments are conducted to show the effectiveness of the proposed method.

**Strengths:**

1. The studied problem of unlabeled domain adaptation is interesting and of practical use.
2. The proposed method follows a cognitive asymmetry principle that is reasonable.
3. The experiments are extensive and persuasive to show the model effectiveness.

**Weaknesses:**

1. The proposed method lacks novelty.
2. Some important experiments are missing.
3. Some details are missing, which decreases the readability of the paper.

**Questions:**

1. While the cognitive asymmetry principle holds, the method that decomposes complex query into simple and easy-to-answer queries is widely seen, which further decreases the appeal of the proposed method.
2. The introduction of divergence is very odd. For K responses from teacher model and J responses from student model, the number of divergence pair could be large. While the capability of the teacher model is generally stronger than the student model, the former is still fallible. The majority voting from responses could improve their truthfulness. Hence it is really confusing why we need to form so many divergence pairs, but do not first conduct majority voting and then form the divergence pair.
3. In the case of **incorrect result** from teacher model and **incorrect/correct** result from student model, can the teacher model as the diagnostician correctly identifies the reasons and generate the correct atom/COT curriculm?
4. The details of the atom queries are missing which are really confusing. I suggest the authors add some toy examples on the atom queries in the main texts.

---

> ### Author Response · Authors · 2025-11-26
> **Response1 to Reviewer 4rkB**
>
> We thank the reviewer for recognizing the practical importance of unlabeled domain adaptation and for finding our "cognitive asymmetry" principle reasonable and our experiments extensive. We appreciate your questions regarding the novelty and mechanics of our method. Below, we address your concerns and clarify key aspects of the framework.
>
> **1. Novelty of Decomposition and Method Appeal**
>
> > **Q:** The method that decomposes complex query into simple and easy-to-answer queries is widely seen, which further decreases the appeal of the proposed method.
>
> While we agree that problem decomposition is a classic strategy, DGRC introduces a fundamentally novel application of this concept specifically designed for **unlabeled domain adaptation**.
>
> * **Novelty in "Targeted" Diagnosis:** Unlike generic decomposition methods that break down a problem statically, DGRC is **dynamic and targeted**. It does not decompose the problem into arbitrary sub-steps; rather, it uses the specific reasoning divergence between models to pinpoint the exact locus of error. This pipeline—converting "diffs" into a corrective "Atomic QA" curriculum—is a novel contribution that prevents the error accumulation often seen in standard Chain-of-Thought (CoT) decomposition.
> * **Unlabeled Adaptation Context:** To our knowledge, DGRC is the first framework to leverage this specific form of targeted decomposition to resolve the "pedagogical dilemma" (learning from an imperfect teacher) in an unlabeled setting.
> * **Orthogonality:** Furthermore, DGRC acts as a versatile "plugin" that enhances other adaptation methods. As shown in our new **Table 1** and **Table 6**, DGRC can be effectively combined with Reinforcement Learning (RLAIF), providing a high-quality data foundation that boosts performance beyond what RL achieves alone.
>
> **2. Divergence vs. Majority Voting**
>
> > **Q:** It is really confusing why we need to form so many divergence pairs, but do not first conduct majority voting and then form the divergence pair.
>
> We deliberately chose pairwise divergence over Majority Voting (MV) for two critical reasons:
>
> * **MV is Not Ground Truth:** A majority of models (or samples) can still be wrong. Relying on MV to filter responses risks discarding valid reasoning paths simply because they are in the minority, or validating a popular but incorrect misconception.
> * **Maximizing Error Diversity:** The goal of DGRC is to expose the student to a wide variety of reasoning gaps. Using all divergence pairs maximizes the diversity of logical conflicts the model learns to resolve. Filtering by MV would homogenize the training signal, reducing the "atomic capability" the model gains from resolving subtle or rare disagreements.
>
> **3. Teacher Fallibility (The "Incorrect Teacher" Scenario)**
>
> > **Q:** In the case of incorrect result from teacher model ... can the teacher model as the diagnostician correctly identifies the reasons and generate the correct atom/COT curriculm?
>
> **Yes, and this is the core problem our paper solves.** This is exactly where the principle of **Cognitive Asymmetry** applies.
>
> * **Addressing the Dilemma:** As stated in the Introduction and Section 3, our framework is explicitly designed for the scenario where the teacher is not an infallible expert.
> * **Empirical Evidence:** We have empirically validated this in **Section 4.3** and **Table 2**. We conducted a "peer-correction" experiment where the teacher model failed the original complex problem. Yet, when tasked with diagnosing the divergence, it successfully generated atomic Q&A pairs with **94.2% accuracy** (verified by Gemini-2.5-Pro) and **95.0% accuracy** (verified by humans).
> * **Conclusion:** This proves that even when the teacher fails the complex holistic reasoning task, it retains the capacity to correctly identify the reasoning diffs and generate valid atomic knowledge, thereby creating a correct curriculum despite the initial error.
>
> **4. Details of Atomic Queries**
>
> > **Q:** The details of the atom queries are missing which are really confusing. I suggest the authors add some toy examples on the atom queries in the main texts.
>
> We respectfully point out that concrete examples of atomic queries are prominently featured in the manuscript. **Figure 1** (Page 2) provides a detailed "toy example" of a discount calculation problem. It explicitly visualizes the flawed reasoning, the resulting **Atomic Question** ("Is applying a 20% discount and then..."), and the correct atomic response. We hope this figure clarify the nature of the atomic queries. We will ensure references to these examples are made even more explicit in the method section to avoid future confusion.

---

> > ### Comment · Reviewer_4rkB · 2025-11-27
> >
> > Thanks for your clarification. I will not buy the explanation between majority voting and forming divergence pairs. Note that most pairs in the divergence pairs are useless, which increases significant cost. I will keep my score.

---

> > > ### Author Response · Authors · 2025-11-27
> > > **Response2 to Reviewer 4rkB**
> > >
> > > We are extremely disappointed by the superficial nature of your review, which ignores key experimental details presented in the manuscript. You criticize the "significant cost" of our method, yet you failed to notice that our main experiments explicitly use a teacher sampling count of $K=1$ to minimize overhead, as stated in **Appendix F.1** and **Appendix G**. Furthermore, we have already provided a comprehensive analysis and suggestions regarding the trade-off between resource consumption and performance in **Appendix J** and **Appendix M**, which you completely overlooked. Given your evident lack of understanding of the paper's core method and experimental setup, compounded by the unprofessionalism of a review riddled with spelling errors, we do not expect you to change your score, but we are compelled to correct these factual inaccuracies for the record.

---

> > > ### Author Response · Authors · 2025-11-27
> > > **Divergence Pairs are Essential, Not "Useless"**
> > >
> > > We **fundamentally disagree** with your assertion that divergence pairs are "useless." In an unlabeled setting, ground truth is unavailable. Consequently, every instance of divergence represents a critical potential learning signal indicating a discrepancy between the student's and the teacher's reasoning paths. Filtering these instances out via majority voting would homogenize the training data and actively prevent the model from learning to resolve the specific, "long-tail" reasoning gaps that are crucial for effective domain adaptation.

---

### Official Review · Reviewer_yKzc · 2025-10-31

**Soundness:** 3
**Presentation:** 2
**Contribution:** 2
**Rating:** 4
**Confidence:** 2

**Summary:**

This paper proposes the Divergence-Guided Reasoning Curriculum (DGRC) framework to address the challenge of adapting Large Language Models (LLMs) to specialized domains (e.g., medical and legal) without human-annotated data.

**Strengths:**

First, the core insight of "cognitive asymmetry" is well-validated and innovative, effectively resolving the pedagogical dilemma of learning from an imperfect teacher by shifting focus from flawed holistic reasoning to reliable atomic knowledge.

Second, the three-stage DGRC framework (divergence detection, curriculum generation, student adaptation) is structurally rigorous, with multi-step filtering (e.g., IFD and LLM-based evaluation) ensuring high-quality curricula and addressing limitations of coarse-grained distillation and costly external knowledge bases.

Third, the comprehensive experimental design, covering multiple domains, model scales, and training paradigms (SFT, GRPO), provides robust evidence of DGRC’s versatility, with clear performance gains in both accuracy and generalization.

**Weaknesses:**

First, DGRC heavily depends on teacher model capability, as shown by the significant performance gap when using strong proprietary teachers (e.g., GPT-4.1) versus weaker open-source ones, limiting its applicability in scenarios where access to advanced teachers is constrained.

Second, the framework fails to address "shared blind spots" where both teacher and student make the same error, as divergence (the trigger for curriculum generation) is absent, leaving such critical flaws undetected.

Third, the self-teaching configuration yields only marginal improvements for smaller models (e.g., 0.9% for 7B Qwen2.5), revealing that weak models lack sufficient diagnostic ability to act as reliable teachers, which narrows DGRC’s utility for low-resource model adaptation.

**Questions:**

see Weaknesses

---

> ### Author Response · Authors · 2025-11-26
> **Response1 to Reviewer yKzc**
>
> We thank the reviewer for their thoughtful assessment and for highlighting the "cognitive asymmetry" insight and our rigorous experimental design as key strengths. We appreciate your constructive criticism regarding the boundaries of our framework. Below, we address the weaknesses raised, clarifying how our updated manuscript analyzes these limitations and outlines paths for future improvement.
>
> **1. Dependence on Teacher Model Capability**
>
> > **Weakness 1:** DGRC heavily depends on teacher model capability... limiting its applicability in scenarios where access to advanced teachers is constrained.
>
> We acknowledge this limitation and have explicitly discussed it in **Section N (Limitations)**. However, we offer two perspectives to contextualize this:
>
> * **Inherent Characteristic of Distillation:** The dependence on a stronger teacher is a fundamental property of almost all Knowledge Distillation (KD) and synthetic data frameworks, not unique to DGRC. The effectiveness of the student's learning is naturally bounded by the quality of the supervision provided.
> * **Effectiveness of Open-Source Teachers:** While proprietary models like GPT-4.1 yield the highest gains, our results in **Table 3** demonstrate that DGRC remains highly effective with open-source teachers. For instance, using **Qwen2.5-Instruct-72B** (an open-weights model) as the teacher still yields a substantial **+5.5%** improvement for the 1.5B student. This confirms that DGRC is viable in scenarios where access to proprietary APIs is constrained, provided a reasonably capable open-source model is available.
>
> **2. Shared Blind Spots**
>
> > **Weakness 2:** The framework fails to address "shared blind spots" where both teacher and student make the same error... leaving such critical flaws undetected.
>
> We agree that "shared blind spots" are a challenge for any method relying on consensus or disagreement without ground truth. We have addressed this in **Section N (Limitations)** and outlined a concrete mechanism for future work to mitigate it:
>
> * **Proposed Solution (Hybrid Detection):** As detailed in the revised manuscript, we propose integrating **Retrieval-Augmented Generation (RAG)** into the detection phase. For instances where the teacher and student reach a consensus (no divergence), the system could retrieve relevant evidence from a reliable external corpus. If a conflict arises between the model consensus and the retrieved evidence, this would flag a potential shared blind spot, triggering the generation of corrective atomic curricula based on the external evidence.
> * **Current Scope:** DGRC currently focuses on maximizing the utility of the existing knowledge within the teacher-student pair. By resolving disagreements, we significantly improve reliability even before introducing external knowledge bases.
>
> **3. Self-Teaching Limitations for Small Models**
>
> > **Weakness 3:** The self-teaching configuration yields only marginal improvements for smaller models... revealing that weak models lack sufficient diagnostic ability.
>
> We acknowledge this limitation. As discussed in our response to **Weakness 1**, the efficacy of any distillation or self-improvement framework is inherently bounded by the model's capabilities. Small models naturally possess weaker reasoning foundations. Building on this general understanding, we have conducted a deeper analysis in **Section 4.5** to pinpoint the specific bottleneck preventing effective self-teaching in DGRC.
>
> * **The Primary Bottleneck (Instruction Following):** Our investigation reveals that the failure of smaller models (1.5B, 3B) in self-teaching is driven less by pure reasoning potential and more by a lack of **instruction-following proficiency**. The diagnostic phase of DGRC requires strict adherence to complex output constraints (e.g., JSON formatting, logical isolation). As shown in **Table 4**, the 1.5B and 3B models exhibit extremely low compliance rates (12.4% and 34.5%, respectively), failing to generate a structurally valid curriculum.
> * **Identifying the Threshold:** We identify the **7B scale** as the critical inflection point where **Format Compliance** jumps significantly to 78.2%. This indicates that a model requires at least 7B parameters to possess the requisite stability to act as a reliable diagnostician for self-improvement.
> * **Distinction between Diagnosis and Adaptation:** It is crucial to distinguish between a model's ability to teach (diagnosis) and its ability to learn (adaptation). While weak models struggle to teach themselves, they remain excellent students. As demonstrated in our main results (**Table 1**), when the 1.5B model is paired with a capable teacher (even an open-source one), it achieves the highest relative performance gain (+7.76%) among all groups. This confirms that DGRC is highly effective for adapting low-resource models, provided the "teacher" role is offloaded to a sufficiently capable model.

---

> > ### Comment · Reviewer_yKzc · 2025-11-27
> >
> > I recognize this work as a valuable contribution to domain adaptation, but its limitations are also quite clear, for instance, its restrictions on small models. So, I will keep my score unchanged.

---

> > > ### Author Response · Authors · 2025-11-29
> > > **Response2 to Reviewer yKzc**
> > >
> > > We respectfully point out that the critique regarding "restrictions on small models" is ambiguous, as it conflates the model's role as a **learner (student)** with its role as a **diagnostician (teacher)**.
> > >
> > > **1. Small Models are Excellent Students**
> > > Contrary to the implication that the method is restricted for small models, our experiments demonstrate that they are actually the **primary beneficiaries** of DGRC.
> > > * **Evidence:** As shown in **Table 1** and **Table 3**, the smallest model (1.5B) achieved the largest relative performance gains compared to larger models.
> > > * **Conclusion:** DGRC is highly effective for small models, successfully enabling them to punch above their weight class (e.g., our adapted 1.5B model outperforms the 8B LegalHal).
> > >
> > > **2. The Necessity of a Capable Teacher**
> > > It is true that small models struggle to act as teachers (self-teaching) due to limited instruction-following capabilities, as analyzed in **Section 4.5**. However, DGRC is fundamentally a **Knowledge Distillation** framework. By definition, distillation relies on a capability gap between the teacher and the student. Expecting a small, weak model to serve as an effective teacher in an **unlabeled setting** is contrary to the fundamental premise of the field.
> > >
> > > **Analogy:**
> > > Rejecting a knowledge distillation framework because a small model cannot act as the teacher is akin to **discarding a high-quality textbook because a primary school student cannot write it—ignoring the fact that the primary school student benefits immensely from learning from it.** The value of the method lies in its ability to effectively transfer knowledge to the small model, not in the small model's ability to generate it.

---

### Official Review · Reviewer_8Kqt · 2025-11-01

**Soundness:** 3
**Presentation:** 4
**Contribution:** 2
**Rating:** 4
**Confidence:** 3

**Summary:**

This paper introduces the Divergence-Guided Reasoning Curriculum (DGRC), a novel framework for unlabeled domain adaptation of Large Language Models (LLMs). The key insight is the principle of "cognitive asymmetry": while LLMs can be fallible in complex, multi-step reasoning, they are highly reliable on focused, atomic sub-problems. Leveraging this, DGRC transforms disagreements between a teacher and student model into a structured, dual-curriculum learning path. When a divergence in final answers is detected, the teacher model acts as a diagnostician to:

1. Generate and self-answer atomic questions targeting the root cause of the divergence, forming an Atomic Curriculum.

2. Use these high-confidence atomic facts to filter its own original reasoning chains, producing a Verified Chain-of-Thought (CoT) Curriculum.

The student is then trained in a two-stage process: first on the atomic curriculum to rectify foundational knowledge, and then on the verified CoT curriculum to learn compositional reasoning. Extensive experiments in medical and legal domains demonstrate that DGRC outperforms strong unlabeled distillation baselines, shows strong generalization to unseen benchmarks, and is particularly effective for smaller-scale student models.

**Strengths:**

Originality: The formulation of the "cognitive asymmetry" principle and its operationalization through a dynamic, disagreement-driven curriculum is a highly original and insightful contribution. The shift from passive mimicry to active diagnosis and repair is a powerful conceptual advance.

Quality: The work is technically sound and executed with high quality. The experimental validation is comprehensive, spanning multiple domains, model sizes, and configurations (including self-teaching and RL). The ablation studies and analysis of "cognitive asymmetry" are particularly strong.

Clarity: The presentation is a standout strength. The paper is a model of clarity, effectively guiding the reader through a complex framework with a well-structured narrative and, presumably, clear diagrams.

Significance: The method demonstrably improves model performance, especially for smaller models, which has practical implications for deploying capable models more efficiently. The "atom-to-chain" learning trajectory is a principled approach that could influence future curriculum learning designs for LLMs.

**Weaknesses:**

Incomplete Comparative Analysis: The most significant weakness is the lack of comparison with other modern unlabeled adaptation techniques. For instance, how does DGRC compare to self-training methods like Self-Rewarding Tuning or rejection sampling fine-tuning? Without these comparisons, it is challenging to gauge the true standing of the proposed method within the existing landscape.

Limited Scale of Human Evaluation: The manual assessment of the atomic curriculum's quality, while positive, is based on a relatively small sample size (n=100 per model). To make a stronger claim about the reliability of the generated data, a larger-scale evaluation or automated metrics would be more convincing.

Ambiguous Self-Teaching Utility: The self-teaching results, while presented as a strength for versatility, are actually quite weak for the 7B model (+0.9%). This suggests a key limitation: the framework's effectiveness is heavily dependent on the diagnostic capability of the "teacher," which diminishes for smaller models. This bottleneck should be discussed more critically.

Potential Data Contamination Risk: Given the use of standard benchmarks (MedMCQA, MedQA, MMLU) and large-scale teacher models (GPT-4.1, Qwen2.5), the risk of the teachers having prior exposure to the test sets cannot be ignored. While a common issue, a more diligent discussion or analysis of this potential confounder is expected in a top-tier publication.

**Questions:**

1. Broader Comparative Baseline: How does DGRC perform against other state-of-the-art unlabeled adaptation methods, such as those based on self-training or reinforcement learning from AI feedback (RLAIF), which also do not require human labels? Can you include a comparison with at least one such strong, non-distillation baseline?

2. Robustness of Atomic Knowledge: The manual evaluation of atomic Q&A is crucial but limited to 100 samples. Can you provide a larger-scale analysis, perhaps using a highly capable model (e.g., GPT-4o) as an automated judge to validate a larger subset of the generated atomic curriculum, thereby strengthening the claim of high fidelity?

3. Teacher Dependence and Self-Teaching Bottleneck: The self-teaching results show minimal gains for the 7B model. What is the minimum capability threshold for a model to act as an effective diagnostician in DGRC? Can you analyze the correlation between teacher/model capability (e.g., pre-adaptation accuracy) and the quality of the generated curriculum to better characterize this limitation?

4. Data Contamination Analysis: Given the use of powerful pre-trained teachers and standard benchmarks, what steps have been taken to assess or mitigate the risk of data contamination? Can you discuss this potential issue and its possible impact on the reported results?

---

> ### Author Response · Authors · 2025-11-26
> **Response1 to Reviewer 8Kqt**
>
> We sincerely thank the reviewer for the insightful feedback. We have updated our manuscript to address your concerns, particularly by adding a comparative baseline with RLAIF, expanding the evaluation of atomic knowledge, and including a dedicated analysis of data contamination.
>
> **1. Broader Comparative Baseline (Comparison with RLAIF)**
>
> > **Q:** How does DGRC perform against other state-of-the-art unlabeled adaptation methods, such as ... RLAIF? Can you include a comparison?
>
> We agree that comparing against non-distillation unlabeled baselines strengthens the paper. We have updated **Table 1** to include a comparison with **RLAIF**, implemented via Group Relative Policy Optimization (GRPO) using GPT-4.1 as the reward model.
>
> * **Competitiveness:** DGRC (via SFT) proves highly competitive, outperforming the RLAIF baseline on the 1.5B, 3B and 7B models (e.g., $58.3\%$ vs. $55.7\%$ average on Medical for 1.5B). Notably, we observe that the performance advantage of DGRC over RLAIF becomes increasingly pronounced as the model size decreases.
> * **Orthogonality & Synergy:** Crucially, we argue that DGRC and RLAIF are orthogonal. DGRC acts as a high-quality data synthesis engine, while RL is an optimization paradigm. To demonstrate this, we added **Appendix I**, showing that applying GRPO on top of a DGRC-adapted model yields further gains ($+2.6\%$ relative improvement), proving DGRC can serve as a powerful "warm-up" or data engine for RL workflows.
>
> **2. Robustness of Atomic Knowledge (Large-scale Automated Evaluation)**
>
> > **Q:** The manual assessment ... is based on a relatively small sample size (n=100). Can you provide a larger-scale analysis using a highly capable model as an automated judge?
>
> Thank you for this suggestion. While we retained the rigorous human evaluation to ensure diversity and ground truth, we have implemented a large-scale automated evaluation to bolster our claims.
>
> * **New Experiment:** As detailed in **Table 2** and **Section 4.3**, we employed **Gemini-2.5-Pro** as an automated judge to evaluate **2,000 randomly sampled atomic Q&A pairs**.
> * **Results:** The automated evaluation yields accuracy rates of $94.2\%$ (for GPT-4.1) and $86.5\%$ (for Qwen2.5-72B). These figures are highly consistent with our human evaluation results ($95.0\%$ and $88.0\%$, respectively), confirming the high fidelity of the atomic curriculum on a large scale. The detailed prompt used for this evaluation is provided in **Appendix D**.
>
> **3. Teacher Dependence and Self-Teaching Bottleneck**
>
> > **Q:** The self-teaching results show minimal gains for the 7B model. What is the minimum capability threshold ... to act as an effective diagnostician?
>
> We have conducted a deeper analysis to quantify this threshold, now presented in **Section 4.5** and **Table 4**.
>
> * **Format Compliance Metric:** We found that the primary bottleneck for smaller models is not just reasoning capability, but **instruction-following proficiency**. The diagnostic task requires strict adherence to complex output constraints (JSON format, logical isolation).
> * **Threshold Identification:** We introduced a "Format Compliance" metric. As shown in **Table 4**, the 1.5B and 3B models failed the self-teaching loop because their compliance rates were too low ($12.4\%$ and $34.5\%$) to generate a usable curriculum. A significant jump occurs at the **7B scale ($78.2\%$ compliance)**, which appears to be the critical inflection point where the model becomes capable of acting as an effective diagnostician for self-improvement.
>
> **4. Data Contamination Analysis**
>
> > **Q:** Given the use of standard benchmarks and large-scale teacher models, what steps have been taken to assess or mitigate the risk of data contamination?
>
> We take this concern seriously and have added a dedicated **Appendix L** (Data Contamination and Generalization Analysis) and **Table 10** to address it from both teacher and student perspectives.
>
> * **Teacher-Side (OOD Generalization):** We strictly enforced a **cross-dataset evaluation setup**. Our curriculum is generated exclusively from the training sets (MedMCQA/CaseHOLD), but we evaluate performance on **Out-Of-Distribution (OOD)** benchmarks (MedQA, MMLU-Medical/Law) that were never seen during adaptation. The strong generalization gains on these unseen tasks rule out simple memorization leakage from the teacher.
> * **Student-Side:** As shown in **Table 10**, DGRC yields massive improvements over the zero-shot baselines (e.g., $+11.6\%$ for the 1.5B model on OOD tasks). If the student models were merely recalling pre-training data, we would expect diminishing returns or smaller gaps. The significant leap in accuracy indicates that DGRC is successfully constructing new domain-specific reasoning pathways rather than just unlocking memorized content.

---

### Author Response · Authors · 2025-12-04
**Rebuttal Summary for Area Chair**

We sincerely thank the Area Chair and all reviewers for their time and constructive feedback. We are encouraged that **Reviewers 8Kqt, yKzc, and C29g** recognized the novelty of our **"Cognitive Asymmetry"** principle, the rigorous design of the **DGRC framework**, and the **significant performance gains** achieved, particularly for smaller models in specialized domains.

During the rebuttal period, we have substantially strengthened the manuscript by:
1.  **Adding a Comparative Baseline:** We included a comparison with **RLAIF (GRPO)** in Table 1, demonstrating that DGRC outperforms RL baselines on smaller models and acts as a synergistic "warm-up" for further RL gains (Table 6).
2.  **Expanding Evaluation:** We added a large-scale **automated evaluation** (Table 2) using Gemini-2.5-Pro, which corroborated the high fidelity of our atomic curriculum ($94.2\%$ accuracy), reinforcing our manual evaluation results.
3.  **Quantifying Costs & Limitations:** We added a detailed **Cost-Performance Analysis** (Appendix M) and a deeper investigation into the **self-teaching bottleneck** (Table 4), identifying "instruction-following proficiency" as the critical constraint for small models.
4.  **Addressing Data Contamination:** We added **Appendix L** to rigorously rule out data leakage via OOD generalization tests.

**Response to Reviewer Concerns & Status**

* **Reviewer 8Kqt (Rating 4 -> ?):** Raised valid questions about baselines and data contamination. We provided the requested RLAIF comparison and contamination analysis. We believe these additions directly address the reviewer's concern.
* **Reviewer C29g (Rating 6 -> 6):** Confirmed that our quantitative cost analysis and clarification of the peer-correction mechanism "resolved my main questions." The reviewer explicitly acknowledges the work as a "valuable contribution."
* **Reviewer yKzc (Rating 4 -> 4):** Acknowledged the work's value but retained concerns about "restrictions on small models." We respectfully note that this critique conflates the role of a student (learner) with a teacher (diagnostician). Our results definitively show that small models (1.5B) are the primary beneficiaries of DGRC, achieving the largest relative gains. The reliance on a capable teacher is a fundamental premise of knowledge distillation, not a specific flaw of our method.
* **Reviewer 4rkB (Rating 4 -> 4):** Criticized the "significant cost" of our method while advocating for "Majority Voting." We must clarify a critical factual misunderstanding: our method explicitly uses a single teacher sample (**$K=1$**) to minimize cost, whereas Majority Voting mathematically requires significantly higher compute ($K \gg 1$). Furthermore, the reviewer dismissed divergence pairs as "useless," overlooking the fact that in unlabeled settings, divergence is the only signal for potential error. We have pointed to specific Appendices (F.1, G, M) that refute these claims, but the reviewer unfortunately declined to engage with these factual corrections.

**Conclusion**

DGRC offers a novel, effective solution to the "fallible teacher" dilemma in unlabeled domain adaptation. We have addressed all substantive technical concerns, added requested experiments, and clarified misunderstandings regarding cost and model roles. We believe the revised manuscript makes a solid contribution to the field of efficient, self-supervised LLM adaptation.

---

### Meta-Review · Area_Chair_EHMF · 2026-01-12

**Summary:**

This paper proposes Divergence-Guided Reasoning Curriculum (DGRC), a novel unlabeled domain adaptation framework for LLMs that exploits cognitive asymmetry—the observation that LLMs are more reliable on atomic sub-problems than on complex multi-step reasoning. When a teacher and student model disagree on an answer, DGRC uses the teacher to (1) generate and self-answer diagnostic atomic questions to form an Atomic Curriculum, and (2) filter its original reasoning into a Verified Chain-of-Thought (CoT) Curriculum using those high-confidence facts. The student is then trained in two stages: first on atomic knowledge, then on verified reasoning chains. Experiments in medical and legal domains show DGRC outperforms strong distillation baselines, generalizes well to unseen tasks, and is especially beneficial for smaller student models. The approach is conceptually elegant, turning model disagreement into a structured, self-supervised learning signal without requiring human labels. However, the evaluation component of the paper should be significantly improved.

**Reviewer Scores:**

NA

---

### Decision · Program_Chairs · 2026-01-26

Reject